# On the Safety of Interpretable Machine Learning: A Maximum Deviation Approach

## Abstract

Interpretable and explainable machine learning has seen a recent surge of interest. We posit that safety is a key reason behind the demand for explainability. To explore this relationship, we propose a mathematical formulation for assessing the safety of supervised learning models based on their maximum deviation over a certification set. We then show that for interpretable models including decision trees, rule lists, generalized linear and additive models, the maximum deviation can be computed exactly and efficiently. For tree ensembles, which are not regarded as interpretable, discrete optimization techniques can still provide informative bounds. For a broader class of piecewise Lipschitz functions, we repurpose results from the multi-armed bandit literature to show that interpretability produces tighter (regret) bounds on the maximum deviation compared with black box functions. We perform experiments that quantify the dependence of the maximum deviation on model smoothness and certification set size. The experiments also illustrate how the solutions that maximize deviation can suggest safety risks.

## 1 Introduction

Interpretable and explainable machine learning (ML) has seen a recent surge of interest because it is viewed as one of the key pillars in making models trustworthy, with implications on fairness, reliability, and safety (Varshney, 2019). It is generally accepted that the ultimate measure of ML explainability is whether a human finds the explanations useful (Doshi-Velez & Kim, 2017; Dhurandhar et al., 2017). However, less attention has been paid to deeper reasons behind the human desire for explainability. In this paper, we posit an important reason is toward achieving *safety* and preventing unexpected harms (Varshney & Alemzadeh, 2017). This reason is implicit in the dichotomy between directly interpretable models vs. post hoc explanations of black-box models. Some argue that only directly interpretable models should be used in high-risk applications (Rudin, 2019). The crux of this argument is that post hoc explanations leave a gap between the explanation and the model that is producing the predictions. Thus, unusual data points may appear to be harmless based on the explanation, but truly cause havoc. This argument however does not explicitly address the question: What does safety mean for such models, and how is it intertwined with interpretability?

Towards answering this question, we propose a mathematical definition for assessing the safety of supervised learning (i.e. predictive) models. Viewing these models as functions mapping an input space to an output space, a key way in which these models can cause harm is through grossly unexpected outputs, corresponding to inputs that are poorly represented in training data. Accordingly, we approach safety assessment for a model by determining its *maximum deviation* over a *certification set* from the output of a *reference model*. The idea of a certification set is that it is a large subset of the input space and is intended to cover all conceivable inputs to the model. The reference model could be a simple, well-understood model or an existing model that has been "tried and tested." These concepts are discussed further in Section 2.

In Section 4, we discuss the computation of the maximum deviation for different model classes and show how this is facilitated by interpretability. For model classes regarded as interpretable, including trees, rule lists, generalized linear and additive models, the maximum deviation can be computed exactly and efficiently by exploiting the model structure. For tree ensembles, which are not regarded as interpretable, discrete optimization techniques can exploit their composition in terms of trees

to provide anytime bounds on the maximum deviation. The case of trees is also generalized in a different direction by considering a broader class of functions that are piecewise Lipschitz, which we argue cover many popular interpretable functions. Here we show that the benefit of interpretability is significantly tighter regret bounds on the maximum deviation compared with black box functions, by appropriately repurposing results from the multi-armed bandit literature in this context. On the other hand, it is less clear that post hoc explanations, which approximate a model locally (Ribeiro et al., 2016; Lundberg & Lee, 2017; Dhurandhar et al., 2018) or globally (Buciluǎ et al., 2006; Hinton et al., 2015), can help with evaluating the maximum deviation and hence safety.

We conduct experiments to illustrate the deviation maximization methods in Section 4 for decision trees, linear and additive models, and tree ensembles. The results in Section 5 quantify how the maximum deviation increases as the size of the certification set increases and as the smoothness of the models decreases. For tree ensembles, we find that the obtained upper bounds on the maximum deviation are informative, showing that the maximum deviation does *not* increase with the number of trees in the ensemble. We also study the feature combinations that maximize deviation, which can shed light on the sources of extreme model outputs and guide further investigation.

Overall, our discussion suggests that a reason for preferring more interpretable models is that it is easier to assess them for unexpected and potentially unsafe outputs.

## 2  Assessing Safety Through Maximum Deviation

We are given a supervised learning model $f$, which is a function mapping an input feature space $\mathcal{X}$ to an output space $\mathcal{Y}$. We wish to assess the safety of this model by finding its worst-case deviation from a given reference model $f_0 : \mathcal{X} \mapsto \mathcal{Y}$. To do this, we additionally require 1) a measure of deviation $D : \mathcal{Y} \times \mathcal{Y} \mapsto \mathbb{R}_+$, where $\mathbb{R}_+$ is the set of non-negative reals, and 2) a certification set $\mathcal{C} \subseteq \mathcal{X}$ over which the deviation is maximized. Then the problem to be solved is

$$\max_{x \in \mathcal{C}} \ D(f(x), f_0(x)). \tag{1}$$

The deviation is worst-case because the maximization is over all $x \in \mathcal{C}$; further implications of this are discussed in Appendix C.

We view problem (1) as a means toward the goal of evaluating safety. In particular, a large deviation value is not necessarily indicative of a safety risk, as two models may differ significantly for valid reasons. For example, one model may capture a useful pattern that the other does not. What large deviation values do indicate, however, is a (possibly) sufficient reason for further investigation. Hence, the maximizing solutions in (1) (i.e., the $\arg\max$) are of operational interest.

Below we further discuss some elements in this problem formulation.

**Output space $\mathcal{Y}$.** In the case of regression, $\mathcal{Y}$ is the set of reals $\mathbb{R}$ or an interval thereof. In the case of binary classification, while $\mathcal{Y}$ could be $\{0, 1\}$ or $\{-1, +1\}$, these limit the possible deviations to binary values as well ("same" or "different"). Thus to provide more informative results, we take $\mathcal{Y}$ to be the space of real-valued scores that are thresholded to produce a binary label. For example, $y$ could be a predicted probability in $[0, 1]$ or a log-odds ratio in $\mathbb{R}$. Similarly for multi-class classification with $M$ classes, $\mathcal{Y} \subset \mathbb{R}^M$ could be a $M$-dimensional space of real-valued scores. Models that abstain can also be accommodated as noted in Appendix A.

**Reference model $f_0$.** The premise of the reference model is that it should be "safe" above all. The simplest case mathematically is for $f_0$ to be a constant function representing a baseline value, for example zero. More generally, $f_0$ may be a simple model that can be readily grasped by a human, may be validated against domain knowledge, or may be based on a small number of expert-selected features. Such models are common in medical risk assessment, consumer finance, and predicting semiconductor yield. By simple, we mean for example a linear model with 10 non-zero coefficients or a decision tree with 10 leaves. The reference model could also be an existing model that is not necessarily simple but has been extensively tested and deployed. In this case, $f$ could be a new version of the model, trained on more recent data or improved in some fashion, and we wish to evaluate its safety before deploying it in place of $f_0$. In this and more complex settings, $f_0$ may not be globally interpretable, but may be so in local regions. The machinery developed in this work could be applied in these settings as well to assess the safety of $f$ (more discussion in Appendix C).

**Certification set $\mathcal{C}$.** The premise of the certification set is that it contains all inputs that the model might conceivably be exposed to. This may include inputs that are highly improbable but not physically or logically impossible (for example, a severely hypothermic human body temperature of 27°C). Thus, while $\mathcal{C}$ might be based on the support set of a probability distribution or data sample, it does not depend on the likelihood of points within the support. The set $\mathcal{C}$ may also be a strict superset of the training data domain. For example, a model may have been trained on data for males, and we would now like to determine its worst-case behavior on an unseen population of females.

For tabular or lower-dimensional data, $\mathcal{C}$ might be the entire input space $\mathcal{X}$. For non-tabular or higher-dimensional data, the choice $\mathcal{C} = \mathcal{X}$ may be too unrepresentative because the manifold of realistic inputs is lower in dimension. In this case, if we have a dataset $\{x_i\}_{i=1}^n$, one possibility is to use a union of $\ell_p$ balls centered at $x_i$,

$$\mathcal{C} = \bigcup_{i=1}^n \mathcal{B}_r^p[x_i], \qquad \mathcal{B}_r^p[x_i] = \{x \in \mathcal{X} : \|x - x_i\|_p \leq r\}. \tag{2}$$

The set $\mathcal{C}$ is thus comprised of points somewhat close to the $n$ observed examples $x_i$, but the radius $r$ does not have to be "small".

In addition to determining the maximum deviation over the entire set $\mathcal{C}$, maximum deviations over subsets of $\mathcal{C}$ (e.g., different age groups) may also be of interest. For example, Appendix D.1 shows deviation values separately for leaves of a decision tree, which partition the input space.

## 3 RELATED WORK

Our work relates to a number of different technical directions. Varshney & Alemzadeh (2017) and Mohseni et al. (2021) give qualitative accounts suggesting that directly interpretable models are an inherently safe design because humans can inspect them to find spurious elements; in this paper, we attempt to make those qualitative suggestions more quantitative. Furthermore, several other authors have highlighted safety as a goal for interpretability, but without much further development as done here (Otte, 2013; Doshi-Velez & Kim, 2017; Tomsett et al., 2018; Gilpin et al., 2018; Rudin, 2019). Moreover, there is no consensus on how to measure interpretability, which motivates the relationship explored in this paper between interpretability and the ease of evaluating safety.

In the area of ML verification, robustness certification methods aim to provide guarantees that the classification remains constant within a radius $\epsilon$ of an input point, while output reachability is concerned with characterizing the set of outputs corresponding to a region of inputs (Wong & Kolter, 2018; Raghunathan et al., 2018; Huang et al., 2020). Our problem of deviation maximization (1) is more closely related to output reachability. The differences in our work are: 1) we consider *two* models, a model $f$ to be assessed and a reference $f_0$, and are interested in their difference as measured by deviation function $D$; 2) our focus is *global*, over a comprehensive set $\mathcal{C}$, rather than local to small neighborhoods around input points; 3) we study the role of interpretability in safety verification. Moreover, works in robust optimization applied to machine learning minimize the worst-case probability of error, but this worst case is over parameters of $f$ rather than over individual values of $x$ (Lanckriet et al., 2002). Thomas et al. (2019) present a framework where during the model training, a set of safety tests is specified in order to accept or reject the possible solution. The specification of these tests is left to the model designer but the goal of the proposed solution is to provide a reusable paradigm to support safety in ML solutions.

We build on related literature from model robustness and explainability areas that deals specifically with tree ensembles. Kantchelian et al. (2016) seek to find the smallest perturbation of an input instance to 'evade' a classifier using mixed-integer programming (MIP). Optimization formulations are also explored by Parmentier & Vidal (2021) for the purposes of counterfactual explanations. MIP approaches are computationally intensive however. To address this Chen et al. (2019) introduce graph based approaches for verification on trees. Their central idea, which we use, is to discretize verification computations onto a graph constructed from the way leaves intersect. The verification problem is transformed to finding all maximum cliques. Devos et al. (2021) expand on this idea by providing anytime bounds by probing unexplored nodes.

Safety has become a critical issue in reinforcement learning (RL) with multiple works focusing on making RL policies safe (Amodei et al., 2016; Zhu et al., 2019; Inala et al., 2020; Rupprecht et al.,

2020). There are two broad themes (García et al., 2015): (i) a safe and verifiable policy is learned at the outset by enforcing certain constraints, and (ii) post hoc methods are used to identify bad regimes or failure points of an existing policy. Our current proposal is complementary to these works as we focus on the supervised learning setup viewed from the lens of interpretability. Nonetheless, ramifications of our work in the RL context are briefly discussed in Appendix C.

## 4 DEVIATION MAXIMIZATION FOR SPECIFIC MODEL CLASSES

In this section, we discuss approaches to computing the maximum deviation (1) for $f$ belonging to various model classes. The benefit of interpretable model structure is seen in different guises. For decision trees, generalized linear and additive models in Sections 4.1 and 4.2, exact and efficient computation is possible. For tree ensembles in Section 4.3, their composition in terms of trees allows discrete optimization methods to provide anytime bounds. For a general class of piecewise Lipschitz functions in Section 4.4, the application of multi-arm bandit results yields tighter regret bounds on the maximum deviation. The results in Sections 4.1–4.4 also show that intuitive measures of model complexity, such as the number of leaves or pieces or smoothness of functions, have more precise interpretations as well in terms of the complexity of maximizing deviation. On the other hand, for post hoc explanations in Appendix B.5, we believe that more development will be needed for them to help in evaluating maximum deviation.

To develop mathematical results and efficient algorithms, we will sometimes assume that the reference model $f_0$ is from the same class as $f$. We will also sometimes assume that the certification set $\mathcal{C}$ and other sets are Cartesian products. This means that $\mathcal{C} = \prod_{j=1}^d \mathcal{C}_j$, where for a continuous feature $j$, $\mathcal{C}_j = [\underline{X}_j, \overline{X}_j]$ is an interval, and for a categorical feature $j$, $\mathcal{C}_j$ is a set of categories.

### 4.1 TREES

We begin with the case where $f$ and $f_0$ are both decision trees. A decision tree with $L$ leaves partitions the input space $\mathcal{X}$ into $L$ corresponding parts, which we also refer to as "leaves". We consider only non-oblique trees. In this case, each leaf is described by a conjunction of conditions on individual features and is therefore a Cartesian product as defined above. With $\mathcal{L}_l \subset \mathcal{X}$ denoting the $l$th leaf and $y_l \in \mathcal{Y}$ the output value assigned to it, tree $f$ is described by the function

$$f(x) = y_l \quad \text{if } x \in \mathcal{L}_l, \quad l = 1, \ldots, L, \tag{3}$$

and similarly for tree $f_0$ with leaves $\mathcal{L}_{0m}$ and outputs $y_{0m}$, $m = 1, \ldots, L_0$. *Rule lists* can be seen as one-sided trees (Yang et al., 2017; Angelino et al., 2018) and are subsumed in this discussion.

The partitioning of $\mathcal{X}$ by decision trees and their piecewise-constant nature simplify the computation of the maximum deviation (1). Specifically, the maximization can be restricted to pairs of leaves $(l, m)$ for which the intersection $\mathcal{L}_l \cap \mathcal{L}_{0m} \cap \mathcal{C}$ is non-empty. The intersection of two leaves $\mathcal{L}_l \cap \mathcal{L}_{0m}$ is another Cartesian product, and we assume that it is tractable to determine whether $\mathcal{C}$ intersects a given Cartesian product (see examples in Appendix B.1).

For visual representation and later use in Section 4.3, it is useful to define a bipartite graph, with $L$ nodes representing the leaves $\mathcal{L}_l$ of $f$ on one side and $L_0$ nodes representing the leaves $\mathcal{L}_{0m}$ of $f_0$ on the other. Define the edge set $\mathcal{E} = \{(l, m) : \mathcal{L}_l \cap \mathcal{L}_{0m} \cap \mathcal{C} \neq \emptyset\}$; clearly $|\mathcal{E}| \leq L_0 L$. Then

$$\max_{x \in \mathcal{C}} D(f(x), f_0(x)) = \max_{(l,m) \in \mathcal{E}} D(y_l, y_{0m}). \tag{4}$$

We summarize the complexity of deviation maximization for decision trees as follows.

**Proposition 1.** *Let $f$ and $f_0$ be decision trees as in* (3) *with $L$ and $L_0$ leaves respectively, and $\mathcal{E}$ be the bipartite edge set of leaf intersections defined above. Then the maximum deviation* (1) *can be computed with $|\mathcal{E}|$ evaluations as shown in* (4).

### 4.2 LINEAR AND ADDITIVE MODELS

In this subsection, we assume that $f$ is a generalized additive model (GAM) given by

$$f(x) = g^{-1}\left(\sum_{j=1}^d f_j(x_j)\right), \tag{5}$$

where each $f_j$ is an arbitrary function of feature $x_j$. In the case where $f_j(x_j) = w_j x_j$ for all continuous features $x_j$, where $w_j$ is a real coefficient, (5) is a generalized linear model (GLM). We discuss the treatment of categorical features in Appendix B.2. The invertible link function $g : \mathbb{R} \mapsto \mathbb{R}$ is furthermore assumed to be monotonically increasing. This assumption is satisfied by common GAM link functions: identity, logit ($g(y) = \log(y/(1-y))$), and logarithmic.

Equation (5) implies that $\mathcal{Y} \subset \mathbb{R}$ and the deviation $D(y, y_0)$ is a function of two scalars $y$ and $y_0$. For this scalar case, we make the following intuitively reasonable assumption throughout the subsection.

**Assumption 1.** *For $y, y_0 \in \mathcal{Y} \subseteq \mathbb{R}$, 1) $D(y, y_0) = 0$ whenever $y = y_0$; 2) $D(y, y_0)$ is monotonically non-decreasing in $y$ for $y \geq y_0$ and non-increasing in $y$ for $y \leq y_0$.*

Our approach is to exploit the additive form of (5) by reducing problem (1) to the optimization

$$\max_{x \in \mathcal{S}} \sum_{j=1}^{d} f_j(x_j), \tag{6}$$

for different choices of $\mathcal{S} \subset \mathcal{X}$ and where minimization is obtained by negating all $f_j$. We discuss below how this can be done for two types of reference model $f_0$: decision tree (which includes the constant case $L_0 = 1$) and additive. For the first case, we prove the following result in Appendix B.2:

**Proposition 2.** *Let $f$ be a GAM as in (5) and $\mathcal{S}$ be a subset of $\mathcal{X}$ where $f_0(x) \equiv y_0$ is constant. Then if Assumption 1 holds,*

$$\max_{x \in \mathcal{S}} D(f(x), f_0(x)) = \max \left\{ D\left(g^{-1}\left(\max_{x \in \mathcal{S}} \sum_{j=1}^{d} f_j(x_j)\right), y_0\right), D\left(g^{-1}\left(\min_{x \in \mathcal{S}} \sum_{j=1}^{d} f_j(x_j)\right), y_0\right) \right\}.$$

**Tree-structured $f_0$.** Since $f_0$ is piecewise constant over its leaves $\mathcal{L}_{0m}$, $m = 1, \ldots, L_0$, we take $\mathcal{S}$ to be the intersection of $\mathcal{C}$ with each $\mathcal{L}_{0m}$ in turn and apply Proposition 2. The overall maximum is then obtained as the maximum over the leaves,

$$\max_{x \in \mathcal{C}} D(f(x), f_0(x)) = \max_{m=1,\ldots,L_0}$$

$$\max \left\{ D\left(g^{-1}\left(\max_{x \in \mathcal{L}_{0m} \cap \mathcal{C}} \sum_{j=1}^{d} f_j(x_j)\right), y_{0m}\right), D\left(g^{-1}\left(\min_{x \in \mathcal{L}_{0m} \cap \mathcal{C}} \sum_{j=1}^{d} f_j(x_j)\right), y_{0m}\right) \right\}. \tag{7}$$

This reduces (1) to solving $2L_0$ instances of (6).

**Additive $f_0$.** For this case, we make the additional assumption that the link function $g$ in (5) is the identity function, as well as Assumption 2 below. The implication of these assumptions is discussed in Appendix B.2.

**Assumption 2.** $D(y, y_0) = D(y - y_0)$ *is a function only of the difference $y - y_0$.*

Then $f_0(x) = \sum_{j=1}^{d} f_{0j}(x_j)$ and the difference $f(x) - f_0(x)$ is also additive. Using Assumptions 2, 1 and a similar argument as in the proof of Proposition 2, the maximum deviation is again obtained by maximizing and minimizing an additive function, resulting in two instances of (6) with $\mathcal{S} = \mathcal{C}$:

$$\max_{x \in \mathcal{C}} D(f(x), f_0(x)) = \max \left\{ D\left(\max_{x \in \mathcal{C}} \sum_{j=1}^{d} f_j(x_j) - f_{0j}(x_j)\right), D\left(\min_{x \in \mathcal{C}} \sum_{j=1}^{d} f_j(x_j) - f_{0j}(x_j)\right) \right\}.$$

**Computational complexity of (6).** For the case of nonlinear additive $f$, we additionally assume that $\mathcal{C}$ is a Cartesian product. It follows that $\mathcal{S} = \prod_{j=1}^{d} \mathcal{S}_j$ is a Cartesian product (see Appendix B.2 for the brief justification) and (6) separates into one-dimensional optimizations over $\mathcal{S}_j$,

$$\max_{x \in \mathcal{S}} \sum_{j=1}^{d} f_j(x_j) = \sum_{j=1}^{d} \max_{x_j \in \mathcal{S}_j} f_j(x_j). \tag{8}$$

The computational complexity of (8) is thus $\sum_{j=1}^{d} C_j$, where $C_j$ is the complexity of the $j$th one-dimensional optimization. We discuss different cases of $C_j$ in Appendix B.2; the important point is that the overall complexity is linear in $d$.

In the GLM case where $\sum_{j=1}^{d} f_j(x_j) = w^T x$, problem (6) is simpler and it is less important that $\mathcal{C}$ be a Cartesian product. In particular, if $\mathcal{C}$ is a convex set, so too is $\mathcal{S}$ (again see Appendix B.2 for justification). Hence (6) is a convex optimization problem.

### 4.3 TREE ENSEMBLES

We now extend the idea used for single decision trees in Section 4.1 to tree ensembles. This class covers several popular methods such as Random Forests and Gradient Boosted Trees. It can also cover *rule* ensembles (Friedman & Popescu, 2008; Dembczyński et al., 2010) as a special case, as explained in Appendix B.3. We assume $f$ is a tree ensemble consisting of $K$ trees and $f_0$ is a single decision tree. Let $\mathcal{L}_{l_k}$ denote the $l$th leaf of the $k$th tree in $f$ for $l = 1, \ldots, L_k$, and $\mathcal{L}_{0m}$ be the $m$th leaf $f_0$, for $m = 1, \ldots, L_0$. Correspondingly let $y_{l_k}$ and $y_{0m}$ denote the prediction values associated with each leaf.

Define a graph $\mathcal{G}(\mathcal{V}, \mathcal{E})$, where there is a vertex for each leaf in $f$ and $f_0$, i.e.

$$\mathcal{V} = \{l_k | \forall k = 1, \ldots, K, \, l = 1, \ldots, L_k\} \cup \{m | m = 1, \ldots, L_0\}. \tag{9}$$

Construct an edge for each overlapping pair of leaves in $\mathcal{V}$, i.e.

$$\mathcal{E} = \{(i, j) | \mathcal{L}_i \cap \mathcal{L}_j \neq \emptyset, \, \forall (i, j) \in V, \, i \neq j\}. \tag{10}$$

This graph is a $K + 1$-partite graph as leaves within an individual tree do not intersect and are an independent set. Denote $M$ to be the adjacency matrix of $\mathcal{G}$. Following Chen et al. (2019), a maximum clique $S$ of size $K + 1$ on such a graph provides a discrete region in the feature space with a computable deviation. A clique is a subset of nodes all connected to each other; a maximum clique is one that cannot be expanded further by adding a node. The model predictions $y_c$ and $y_{0c}$ can be ensembled from leaves in $S$. Denote $D(S)$ to be the deviation computed from the clique $S$. Maximizing over all such cliques solves (1). However, complete enumeration is expensive, so informative bounds, either using the merge procedure in Chen et al. (2019) or the heuristic function in Devos et al. (2021) can be used. We use the latter which exploits the $K + 1$-partite structure of $\mathcal{G}$.

Specifically, we adapt the anytime bounds of Devos et al. (2021) as follows. At each step of the enumeration procedure, an intermediate clique $S$ contains selected leaves from trees in $[1, \ldots, k]$ and unexplored trees in $[k + 1, \ldots, K + 1]$. For each unexplored tree, we select a valid candidate leaf that maximizes deviation, i.e.

$$v_k = \underset{l_k, \, l_k \cap i \neq \emptyset, \, \forall i \in S}{\arg \max} D(S \cup l_k). \tag{11}$$

Using these worst-case leaves, a heuristic function

$$H(S) = D(S') = D(S \bigcup_{m=k+1}^{K+1} v_m) \tag{12}$$

provides an upper (dual) bound. In practice, this dual bound is tight and therefore very useful during the search procedure to prune the search space. Each $K + 1$ clique provides a primal bound, so the search can be terminated early before examining all trees if the dual bound is less than the primal bound. We adapt the search procedure from Mirghorbani & Krokhmal (2013) to include the pruning arguments. Appendix B.3 presents the full algorithm. Starting with an empty clique, the procedure adds a single node from each tree to create an intermediate clique. If the size of the clique is $K + 1$ the primal bound is updated. Otherwise, the dual bound is computed. A node compatibility vector is used to keep track of all feasible additions. When the search is terminated at any step, the maximum deviation is bounded by $(D_{lb}, D_{ub})$.

The algorithm works for the entire feature space. When the certification set $\mathcal{C}$ is a union of balls as in Eq. (2), some additional considerations are needed. First, we can disregard leaves that do not intersect with $\mathcal{C}$ during the graph construction phase. A validation step to ensure that the leaves of a clique all intersect with the same ball in $\mathcal{C}$ is also needed.

### 4.4 PIECEWISE LIPSCHITZ FUNCTIONS

We saw the benefits of having specific (deterministic) interpretable functions as well as their extensions in the context of safety. Now consider a richer class of functions that may also be randomized with finite variance. In this case let $f$ and $f_0$ denote the mean values of the learned and reference functions respectively. We consider the case where each function is either interpretable or black box, where the latter implies that query access is the only realistic way of probing the model. This leads to three cases where either both functions are black box or interpretable, or one is black box. What we care about in all these cases[1] is to find the maximum (and minimum) of a function $\Delta(x) = f(x) - f_0(x)$. Let us consider finding only the maximum of $\Delta$ as the other case is symmetric. Given that $f$ and $f_0$ can be random functions $\Delta$ is also a random function and if $\Delta$ is black box a standard way to optimize it is either using Bayesian Optimization (BO) (Auer, 2002) or tree search type bandit methods (Bubeck et al., 2011; Carpentier & Valko, 2015).We repurpose some of the results from this latter literature in our context showcasing the benefit of interpretability from a safety standpoint. To do this we first define relevant terms.

**Definition 1** (Simple Regret (Bubeck et al., 2011)). *If $f_{\mathcal{C}}^*$ denotes the optimal value of the function $f$ on the certification set $\mathcal{C}$, then the simple regret $r_q^{\mathcal{C}}$ after querying the $f$ function $q$ times and obtaining a solution $x_q$ is given by, $r_q^{\mathcal{C}}(f) = f_{\mathcal{C}}^* - f(x_q)$.*

**Definition 2** (Order $\beta$ c-Lipschitz). *Given a (normalized) metric $\ell$ a function $f$ is c-Lipschitz continuous of order $\beta > 0$ if for any two inputs $x$, $y$ and for $c > 0$ we have, $|f(x) - f(y)| \leq c \cdot \ell(x, y)^\beta$.*

**Definition 3** (Near optimality dimension (Bubeck et al., 2011)). *If $\mathcal{N}(\mathcal{C}, \ell, \epsilon)$ is the maximum number of $\epsilon$ radius balls one can fit in $\mathcal{C}$ given the metric $\ell$ and $\mathcal{C}_\epsilon = \{x \in \mathcal{C} | f(x) \geq f_{\mathcal{C}}^* - \epsilon\}$, then for $c > 0$ the c-near optimality dimension is given by, $\upsilon = \max\left(\limsup_{\epsilon \to 0} \frac{\ln \mathcal{N}(\mathcal{C}_{c\epsilon}, \ell, \epsilon)}{\ln(\epsilon^{-1})}, 0\right)$.*

Intuitively, simple regret measures the deviation between our current best and the optimal solution. The Lipschitz condition bounds the rate of change of the function. Near optimality dimension measures the set size for which the function has close to optimal values. The lower the value of $\upsilon$, the easier it is to find the optimum. We now define what it means to have an interpretable function.

**Assumption 3** (Characterizing an Interpretable Function). *If a function $f$ is interpretable, then we can (easily) find $1 \leq m \ll n$ partitions $\{\mathcal{C}^{(1)}, ..., \mathcal{C}^{(m)}\}$ of the certification set $\mathcal{C}$ such that the function $f^{(i)} = \{f(x) | x \in \mathcal{C}^{(i)}\} \ \forall i \in \{1, ..., m\}$ in each partition is c-Lipschitz of order $\beta$.*

*Note that the (interpretable) function overall does not have to be c-Lipschitz of bounded order, rather only in the partitions.* This assumption is motivated by observing different interpretable functions. For example, in the case of decision trees the $m$ partitions could be its leaves, where typically the function is a constant in each leaf ($c = 0$). For rule lists as well a fixed prediction is usually made by each rule. For a linear function one could consider the entire input space (i.e. $m = 1$), where for bounded slope $\alpha$ the function would also satisfy our assumption ($c = \alpha$ and $\beta = 1$). Examples of models that are not piecewise constant or globally Lipschitz are oblique decision trees (Murthy et al., 1994), regression trees with linear functions in the leaves, and functional trees. Moreover, $m$ is likely to be small so that the overall model is interpretable (viz. shallow trees or small rules). With the above definitions and Assumption 3 we now provide the simple regret for the function $\Delta$.

**1. Both black box models:** If both $f$ and $f_0$ are black box then it seems no gains could be made in estimating the maximum of $\Delta$ over standard results in bandit literature. Hence, using Hierarchical Optimistic Optimization (HOO) with assumptions such as $\mathcal{C}$ being compact and $\Delta$ being weakly Lipschitz (Bubeck et al., 2011) with near optimality dimension $\upsilon$ the simple regret after $q$ queries is:

$$r_q^{\mathcal{C}}(\Delta) \leq O\left(\left(\frac{\ln(q)}{q}\right)^{\frac{1}{\upsilon+2}}\right) \tag{13}$$

**2. Both interpretable models:** If both $f$ and $f_0$ are interpretable, then for each function based on Assumption 3 we can find $m_1$ and $m_0$ partitions of $\mathcal{C}$ respectively where the functions are $c_1$ and $c_0$-Lipschitz of order $\beta_1$ and $\beta_0$ respectively. Now if we take non-empty intersections of these partitions where we could have a maximum of $m_1 m_0$ partitions, the function $\Delta$ in these partitions would be $c = 2\max(c_0, c_1)$-Lipschitz of order $\beta = \min(\beta_0, \beta_1)$ as stated next (proof in appendix).

---

[1] For simplicity assume $D(., .)$ to be the identity function.

**Proposition 3.** *If functions $h_0$ and $h_1$ are $c_0$ and $c_1$ Lipschitz of order $\beta_0$ and $\beta_1$ respectively, then the function $h = h_0 - h_1$ is c-Lipschtiz of order $\beta$, where $c = 2\max(c_0, c_1)$ and $\beta = \min(\beta_0, \beta_1)$.*

Given that $\Delta$ is smooth in these partitions with underestimated smoothness of order $\beta$, the simple regret after $q_i$ queries in the $i^{\text{th}}$ partition $\mathcal{C}^{(i)}$ with near optimality dimension $\upsilon_i$ based on HOO is: $r_{q_i}^{\mathcal{C}^{(i)}}(\Delta) \leq O\left(\frac{1}{q_i^{\frac{1}{\upsilon_i + 2}}}\right)$, where $\upsilon_i \leq \frac{d}{\beta}$. If we divide the overall query budget $q$ across the $\pi \leq m_0 m_1$ non-empty partitions equally, then the bound will be scaled by $\pi^{\frac{1}{\upsilon_i + 2}}$ when expressed as a function of $q$. Moreover, the regret for the entire $\mathcal{C}$ can then be bounded by the maximum regret across these partitions leading to the following result:

$$r_q^{\mathcal{C}}(\Delta) \leq O\left(\left(\frac{\pi}{q}\right)^{\frac{\beta}{d+2\beta}}\right) \tag{14}$$

Notice that for a model to be interpretable $m_0$ and $m_1$ are likely to be small (i.e. shallow trees or small rule lists or linear model where $m = 1$) leading to a "smallish" $\pi$ and $\upsilon$ can be much $>> \frac{d}{\beta}$ in case 1. Hence, interpretability significantly reduces the regret in estimating the maximum deviation.

**3. Black box and interpretable model:** Making no further assumptions on the black box model and assuming $\Delta$ satisfies properties mentioned in case 1, the simple regret has the same behavior as equation 13. This is expected as the black box model could be highly non-smooth.

## 5 EXPERIMENTS

We present a case study on the UCI Adult Income dataset (Dua & Graff, 2017), using its given partition into training and test sets. Additional datasets are presented in Appendix D. For the reference model $f_0$, an 8-leaf decision tree (DT) is fit (using scikit-learn's `max_leaf_nodes` parameter) on the training data. This DT has $85.0\%$ accuracy on the test set and is easy for a human to understand and validate (see Figure 3). We take the deviation function $D$ to be the absolute difference between predicted probabilities of the positive (high-income) class. For the certification set $\mathcal{C}$, we consider a union of $\ell_\infty$ balls (2) centered at test set instances ($n = 16281$). The $\ell_\infty$ norm is computed on normalized feature values, where continuous features are standardized and categorical features are one-hot encoded. The case $r = 0$ yields a finite set consisting only of the test set, while $r \to \infty$ corresponds to $\mathcal{C}$ being the entire domain $\mathcal{X}$.

### 5.1 TREES, LINEAR AND ADDITIVE MODELS

We first consider models for which the maximum deviation can be computed exactly: decision trees, generalized linear and additive models. For the first two classes, we use scikit-learn (Pedregosa et al., 2011) to train DTs of varying numbers of leaves (`max_leaf_nodes`) and logistic regression (LR) models with varying amounts of $\ell_1$ regularization. For additive models, we use Explainable Boosting Machines (EBM) from the InterpretML package (Nori et al., 2019) with zero interaction terms. Smoothness is controlled by the `max_bins` parameter, the number of discretization bins for continuous features. Statistics and plots of the resulting models can be found in Appendix D.1.

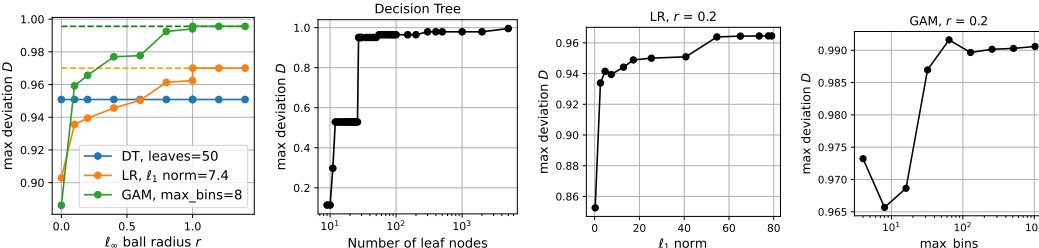

Figure 1: Maximum deviation $D$ for the three models as a function of certification set size (radius $r$, left panel) and model smoothness (number of leaves for DT, $\ell_1$ norm for LR, `max_bins` for GAM).

To evaluate the maximum deviation for DTs, Algorithm 1 is used on a bipartite graph. For the latter two cases where $f$ is generalized additive, $f_0$ is a DT, and radius $r > 0$, we use (7), (6) ($r = 0$ is

handled simply by evaluating the models on the test set). For $r < \infty$ when $\mathcal{C}$ is a union of balls, we maximize separately over each intersection between a ball and a leaf of $f_0$, and take their maximum.

Figure 1 shows a summary of the maximum deviation $D$ as a function of certification set radius $r$ and each model's smoothness parameter; Appendix D.1 has breakdowns by leaves of $f_0$. The first observation is that the deviation can be large even between simple models, namely the 8-leaf reference DT $f_0$, a 50-leaf DT $f$, a sparse LR model (16 nonzeros, $\ell_1$ norm $= 7.4$), and a smooth GAM (see Figure 5 for plots of $f_j$). At the same time, the interpretability of these models allows the deviation to be computed for a range of infinite certification sets with $r > 0$. We can thus quantify the increase in deviation as $r$ increases, as seen in the left of Figure 1. For LR and GAM, $r$ only has to be slightly larger than 1 for the deviation to equal that for $r = \infty$ ($\mathcal{C} = \mathcal{X}$, dashed lines in figure), whereas for DT, the deviation at $r = 0$ is already maximal. The right panels show that the deviation also increases as model smoothness decreases (number of leaves for DT, $\ell_1$ norm for LR, `max_bins` for GAM). The initial increase is rapid for DT and LR. Appendix D.1 reports on the complexity of computing $D$ (running time, number of cliques evaluated).

Studying the solutions that maximize deviation (the $\arg\max$) can bring attention to unanticipated extremes in model output, which may rise to the level of safety risks. Appendix D.1 discusses in detail some maximizing solutions corresponding to Figure 1. A common source of large deviations is the extrapolation of linear and monotonic functions (more generally, unbounded or poorly bounded functions) to extreme points of the certification set, or even parts thereof. At these extreme points, the model $f$ and reference $f_0$ are in conflict, which may be reason for caution or call for domain expert intervention. In the GAM case, the $\arg\max$ also highlights the presence of an unintuitive extreme value in one of the functions $f_j$, which may be an artifact warranting further investigation.

## 5.2 TREE ENSEMBLES

Next, we demonstrate the methods in Section 4.3 on a Random Forest model, where the number of estimators is varied. A time limit of two hours is imposed for the search procedure for each run. There are two sets of experiments, one where the set $\mathcal{C}$ is assumed to be the entire domain, and another where $\mathcal{C}$ is a union of balls.

In Figure 2a we observe that max deviations can be large even for simpler models with fewer estimators. The upper bound on max deviation reduces as the number of estimators increases. The larger number of estimators increases averaging and may serve to make the model smoother. The primal lower bound here is the maximum $D$ for complete $K + 1$-max-cliques. Due to the size of underlying graphs, fewer $K + 1$-max-cliques are evaluated and the lower bound is weak. For safety evaluation, i.e. worst case behavior, the lower bound is of less consequence.

For the same setting, we now consider the case where $\mathcal{C}$ is a union of balls of the test set and the radius $r$ is varied. Figure 2b shows the upper bound on safety as a function of the certification set size for two RF models. As the test set is large in this case, the deviations observed even for small values of $r$ are high and grow to reach the value of the full feature space quickly.

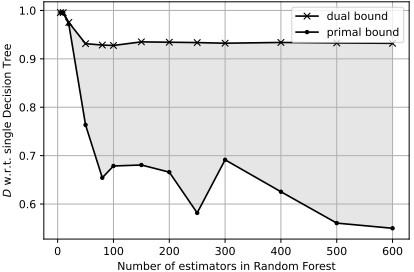

(a) $D$ w.r.t. number of estimators

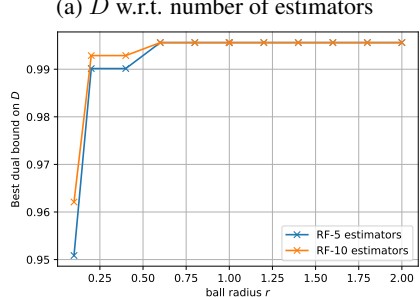

(b) $D$ w.r.t. size of certification set $\mathcal{C}$

Figure 2: Maximum deviation of $f$, a Random Forest, trained on the Adult Income dataset.

## 6 CONCLUSION

We have considered the relationship between interpretability and safety in supervised learning through two main contributions: First, the proposal of maximum deviation as a means toward assessing safety, and second, discussion of approaches to computing maximum deviation and how these are simplified by interpretable model structure. We believe that there is much more to explore in this relationship. Appendix C provides further discussion of several topics.

ETHICS DISCUSSION

The ICLR Code of Ethics requires contributions to avoid harm and contribute to society. The safety of machine learning systems is a fundamental factor in achieving these goals, and has been called out by the European Commission's regulatory framework (High Level Expert Group on Artificial Intelligence, 2020). The commission states seven key dimensions to be evaluated and audited by a cross-disciplinary team: (i) human agency and oversight, (ii) technical robustness and safety, (iii) privacy and data governance, (iv) transparency, (v) diversity, non-discrimination and fairness, (vi) environmental and societal well-being, and (vii) accountability. The second of these dimensions is safety. However, Sloane et al. (2021) argue that algorithmic audits are ill-defined as the underlying definitions are vague. The proposed work helps fill that ill-definedness using a quantitative approach. One may argue against this particular choice of quantification, but it does start the community down the path toward being more concrete in its definitions.

As with many other technologies, the proposed approach may be misused. For example, the reference model may be chosen in a way that hides the safety concerns of the model being evaluated. Transparent documentation and reporting with provenance guarantees can help avoid this kind of purposeful deceit (Arnold et al., 2019).

REPRODUCIBILITY DISCUSSION

To promote reproducibility, all algorithms and mathematical derivations in the paper are presented with a large amount of detail in the main body and appendix. All datasets used in the experiments are publicly-available. All parameters and their settings are fully described.

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

## A    ADDITIONAL PROBLEM FORMULATION DETAILS

**Models that abstain**    The formulation in Section 2 can also accommodate models that abstain from predicting (and possibly defer to a human expert or other fallback system). If $f(x) = \emptyset$, representing abstention, then we may set $D(\emptyset, y_0) = d$ for any $y_0 \in \mathcal{Y}$, where $d > 0$ is an intermediate value less than the maximum value that $D$ can take (Bartlett & Wegkamp, 2008). The value $d$ might also be less than a "typically bad" value for $D$, to reward the model for abstaining when it is uncertain.

## B    ADDITIONAL DETAILS ON DEVIATION MAXIMIZATION FOR SPECIFIC MODEL CLASSES

### B.1    TREES

**Rule lists**    A rule list is a nested sequence of IF-THEN-ELSE statements, where the IF condition usually involves a single feature and the THEN consequent is an output value. As discussed in Yang et al. (2017); Angelino et al. (2018), such rule lists are one-sided trees and are thus subsumed in the discussion of Section 4.1. The number of leaves in the equivalent tree is equal to the number of rules in the list (including the last default rule).

**Intersection of $\mathcal{C}$ with a Cartesian product**    If $\mathcal{C} = \prod_{j=1}^{d} \mathcal{C}_j$ is also a Cartesian product, then determining whether the intersection is non-empty amounts to checking whether all of the coordinate-wise intersections with $\mathcal{C}_j$, $j = 1, \ldots, d$, are non-empty. If $\mathcal{C}$ is not a Cartesian product but is a union of $\ell_\infty$ balls (which are Cartesian products), then the intersection is non-empty if the intersection with any one ball is non-empty.

**Additive reference model**    For the case where $f$ is a decision tree and $f_0$ is a generalized additive model, if the deviation function is symmetric, $D(y, y_0) = D(y_0, y)$, then this case is covered in Section 4.2.

### B.2    LINEAR AND ADDITIVE MODELS

**Categorical features**    A function $f_j(x_j)$ of a categorical feature $x_j$ can be represented in two ways, depending on whether $f$ is a GAM or a GLM. In the GAM case, we may use the native representation in which $x_j$ takes values in a finite set $\mathcal{X}_j$ of categories. In the GLM case, $x_j$ is one-hot encoded into multiple binary-valued features $x_{jk}$, one for each category $k$. Then any function $f_j$ can be represented as a linear function,

$$f_j(x_j) = \sum_{k=1}^{|\mathcal{X}_j|} w_{jk} x_{jk},$$

where $w_{jk}$ is the value of $f_j$ for category $k$.

**Implication of Assumption 1**    The second condition implies that the deviation increases or stays the same as $y$ moves away from $y_0$ in either direction.

*Proof of Proposition 2.* Let $x \in \mathcal{S}$ and $S(x) = \sum_{j=1}^{d} f_j(x_j)$. Under Assumption 1.1, if $S(x) = g(y_0)$, then

$$D(f(x), f_0(x)) = D(g^{-1}(g(y_0)), y_0) = D(y_0, y_0) = 0.$$

As $S(x)$ increases from $g(y_0)$, $f(x)$ also increases because $g^{-1}$ is an increasing function, and $D(f(x), y_0)$ increases or stays the same due to Assumption 1.2. Similarly, as $S(x)$ decreases from $g(y_0)$, $f(x)$ decreases, and $D(f(x), y_0)$ again increases or stays the same. It follows that to maximize $D(f(x), y_0)$, it suffices to separately maximize and minimize $S(x)$, compute the resulting values of $D(f(x), y_0)$, and take the larger of the two. This yields the result. $\qquad\square$

**Implication of Assumption 2 and identity link function** $g$    These two assumptions imply that the deviation is measured on the difference between $f$ and $f_0$ in the space in which they are additive. For example, if $f$ and $f_0$ are logistic regression models predicting the probability of belonging to one of the classes, the difference is taken in the log-odds (logit) domain. It is left to future work to determine other assumptions under which problem (1) is tractable when $f$ and $f_0$ are both additive.

**Cartesian product** $\mathcal{C}$ **implies Cartesian product** $\mathcal{S}$    In the cases of constant and additive $f_0$, $\mathcal{S} = \mathcal{C}$. In the decision tree case, since each leaf is a Cartesian product $\mathcal{L}_{0m} = \prod_{j=1}^{d} \mathcal{R}_{mj}$, the intersections $\mathcal{S} = \mathcal{L}_{0m} \cap \mathcal{C}$ are also Cartesian products $\prod_{j=1}^{d} \mathcal{S}_j$ where $\mathcal{S}_j = \mathcal{R}_{mj} \cap \mathcal{C}_j$.

**One-dimensional optimization complexities** $C_j$    For discrete-valued $x_j$, $C_j$ is proportional to the number of allowed values $|\mathcal{S}_j|$. For continuous $x_j$, it is common to use spline functions or tree ensembles as $f_j$ in constructing GAMs. In the former case, $C_j$ is proportional to the number of knots. In the latter, the tree ensemble can be converted to a piecewise constant function and $C_j$ is then proportional to the number of pieces. Lastly in the case where $f_j(x_j) = w_j x_j$ is linear and $\mathcal{S}_j = [\underline{X}_j, \overline{X}_j]$ is an interval, $C_j = O(1)$ because it suffices to evaluate the two endpoints.

**Convex** $\mathcal{S}$    If $\mathcal{C}$ is a convex set, then in the cases of constant and additive $f_0$, $\mathcal{S} = \mathcal{C}$ is also convex. In the case of tree-structured $f_0$, $\mathcal{S} = \mathcal{L}_{0m} \cap \mathcal{C}$ and each leaf $\mathcal{L}_{0m}$ can be represented as a convex set, with interval constraints on continuous features and set membership constraints on categorical features. The latter can be represented as $x_{jk} = 0$ constraints on the one-hot encoding (see "Categorical features" paragraph above) for non-allowed categories $k$. Hence $\mathcal{S}$ is also convex.

As a specific example, suppose that $\mathcal{S}$ is the product of independent constraints on each categorical feature and an $\ell_p$ norm constraint on the continuous features jointly. The maximization over each categorical feature has complexity $C_j = |\mathcal{S}_j|$ as noted above, while the maximization of $w^T x$ over continuous features lying in an $\ell_p$ ball has closed-form solutions for the common cases $p = 1, 2, \infty$.

### B.3    TREE ENSEMBLES

The full algorithm for clique search from Section 4.3 is presented in Algorithm 1. It uses $Z$ as a node compatibility vector to keep track of valid leaves and $B$ a set of trees/partites not yet covered by the maximum clique. The algorithm starts with and empty clique $S$ and anytime bounds as 0. It starts the search with the smallest tree to limit the search space. This is typically $f_0$. Each leaf is added to the intermediate clique $S$ in turn (Line 6). A stronger primal bound can be achieved if the traversal is ordered in a meaningful way. In particular, starting with nodes with the highest heuristic function value $H(S)$ aids the algorithm to focus on better areas of the search space.

If the size of the clique is $K+1$ the primal bound is updated. Otherwise, the dual bound is computed. If the node is promising, the algorithm recurses to the next level. When the search is terminated at any step, the maximum deviation is bounded by $(D_{lb}, D_{ub})$.

**Rule ensembles**    Similar to the tree ensembles considered in Section 4.3, a rule ensemble is a linear combination of conjunctive rules, where the antecedent is a conjunction of conditions on individual features, and the consequent takes a real value if the antecedent is true and zero otherwise. They are produced by algorithms such as SLIPPER (Cohen & Singer, 1999), that of Rückert & Kramer (2006), RuleFit (Friedman & Popescu, 2008), ENDER (Dembczyński et al., 2010) and have also been referred to as generalized linear rule models (Wei et al., 2019). A rule ensemble can be converted into a tree ensemble by converting each conjunctive rule into an IF-THEN-ELSE rule list, which is a one-sided tree (see Appendix B.1). Specifically, the conditions in the conjunction are taken in any order, each condition is negated to become an IF condition, and the THEN consequents are all output values of zero. The final ELSE consequent, which is reached if all the IF conditions are false (and hence the original rule holds), returns the output value of the original rule. The number of leaves in the resulting tree equals the number of conditions in the conjunction plus one.

---

**Algorithm 1** Max clique search for maximum deviation

---

**Require:** $M$ adjacency matrix, $H$ heuristic function
1: $Z[i] = 1 \forall i \in V$, $B = \{1, 2, \ldots, K+1\}$, $S = \emptyset$      $\triangleright$ All nodes valid, all trees uncovered
2: $Q = $ ENUMERATE$(Z, B, S)$
3: **Initialize:**
     $D_{lb} = 0$, $D_{ub} = 0$      $\triangleright$ Anytime bounds
4: **function** ENUMERATE$(Z, B, S)$
5:     $t = \arg\max_b \{ |Z_b| \mid b \in B \}$      $\triangleright$ Uncovered tree with fewest valid nodes
6:     **for** $i$ in $Z_t$ **do**
7:        $Z[i] = 0$      $\triangleright$ Mark node as incompatiable
8:        $S = S \cup \{i\}$      $\triangleright$ Add to candidate clique
9:        **if** $|S| = K+1$ **then**:      $\triangleright$ Is it a max clique?
10:          $D_{lb} = \max(D_{lb}, D(S))$      $\triangleright$ Update primal bound
11:          $Q = Q \cup S$      $\triangleright$ Add to set of max cliques
12:          $S = S \setminus \{i\}$      $\triangleright$ Backtrack
13:        **else**
14:          $Z_{t+1} = Z_t \wedge M(i)$      $\triangleright$ Update valid nodes
15:          $B = B \setminus \{t\}$      $\triangleright$ Update uncovered trees
16:          $D_{ub} = \max(D_{ub}, H(S))$      $\triangleright$ Update dual bound
17:          **if** $D_{ub} > D_{lb}$ **then**:      $\triangleright$ Pruning by bound
18:            ENUMERATE$(Z_{t+1}, B, S)$      $\triangleright$ Recurse to next level
19:          **end if**
20:          $S = S \setminus \{i\}$      $\triangleright$ Backtrack
21:          $B = B \cup \{t\}$
22:        **end if**
23:     **end for**
24: **end function**

---

### B.4 PIECEWISE LIPSCHITZ FUNCTIONS

*Proof of Proposition 3.* Consider two inputs $x$ and $y$ then,

$$|h(x) - h(y)| = |(h_0 - h_1)(x) - (h_0 - h_1)(y)| = |h_0(x) - h_0(y) + h_1(y) - h_1(x)|$$
$$\leq |h_0(x) - h_0(y)| + |h_1(x) - h_1(y)| \leq c_0 \cdot \ell(x, y)^{\beta_0} + c_1 \cdot \ell(x, y)^{\beta_1}$$
$$\leq c \cdot \ell(x, y)^{\beta}$$

where, $c = 2\max(c_0, c_1)$ and $\beta = \min(\beta_0, \beta_1)$.      $\square$

**Other choices for** $D(.,.)$**:** The results assumed $D(.,.)$ to be the identity function, where $\Delta = D(f_0, f)$. This choice of function clearly satisfies assumptions 1 and 2. Again consistent with these assumptions we look at some other choices for $D(.,.)$. If $D(.,.)$ were an affine function with a positive scaling such as $D(y_0, y) = \alpha(y_0 - y) + b$ where $\alpha > 0$, then our result in equation 14 would be unchanged as only the Lipschitz constant of $\Delta$ would change, but not its (underestimated) order. If the function were a polynomial or exponential however, no such guarantees can be made and we would be back to case 1.

### B.5 POST HOC EXPLANATIONS FOR OTHER MODEL CLASSES

For model classes beyond the ones discussed in Section 4, it is less clear whether there exist reasonably tractable algorithms that guarantee exact computation of or bounds on the maximum deviation. In this case, it is natural to ask whether post hoc explanations for the model can help. One way in which this could occur is if the post hoc explanation approximates the model $f$ by a simpler model $\hat{f}$ and if the deviation function $D$ satisfies the triangle inequality $D(f(x), f_0(x)) \leq D(f(x), \hat{f}(x)) + D(\hat{f}(x), f_0(x))$. Then the maximum deviation in (1) would be bounded as

$$\max_{x \in \mathcal{C}} D(f(x), f_0(x)) \leq \max_{x \in \mathcal{C}} D(f(x), \hat{f}(x)) + \max_{x \in \mathcal{C}} D(\hat{f}(x), f_0(x)). \tag{15}$$

While we may choose $\hat{f}$ to be interpretable so that the rightmost maximization is tractable, the middle maximization asks for a *uniform* bound on the deviation between $f$ and $\hat{f}$, i.e, the fidelity of $\hat{f}$. We are not aware of a post hoc explanation method that provides such a guarantee. Indeed, in general, the middle maximization might not be any easier than the left-hand one that we set out to bound.

A (practical) possibility may be to perform quantile regression (Koenker, 2005) for a large enough quantile to learn $\hat{f}$, as opposed to minimizing expected error as is typically done. This may be an interesting direction to explore in the future as quantile regression algorithms are available for varied model classes including linear models, tree ensembles (Meinshausen, 2006) and even neural networks (Petneházi, 2019). More investigation is needed into whether quantile regression methods can provide approximate guarantees on the middle term in (15).

## C  FURTHER DISCUSSION

**Worst-case approach**   The formulation of (1) as the worst case over a certification set represents a deliberate choice to depend as little as possible on a probability distribution or a dataset sampled from one. As stated in Section 2, Certification Set paragraph, $\mathcal{C}$ can depend at most on (an expanded version of) the support set of a distribution. The reason for this choice is because safety is an out-of-distribution notion: harmful outputs often arise precisely because they were not anticipated in the data. The trade-off inherent in this choice is that the maximum deviation may be more conservative than needed. The high maximum deviation values in e.g. Figures 1 and 2 may reflect this. Given definition (1) as a starting point in this paper, future work could consider variations that depend more on a distribution and are thus less conservative, but may also offer a weaker safety guarantee.

**Choice of reference model**   The proposed definition of maximum deviation (1) depends on the choice of reference model $f_0$. Different choices will lead to different deviation values and, perhaps more importantly, different combinations of features that maximize the deviation. We have discussed possible choices in Section 2, and the results in Section 4 indicate that, as with the assessed model $f$, interpretable forms for $f_0$ can ease the computation of maximum deviation. Beyond these guidelines, it is up to ML practitioners and domain experts to decide on appropriate reference models for their application (and there may be benefit to considering more than one). We mention an additional concern with the reference model in the Ethics Discussion.

For some real applications it may be difficult to come up with a globally interpretable reference model. But specific to particular scenarios it may be possible. For instance, it might be difficult to provide general rules for how to drive a car, but in specific scenarios such as there being an obstacle in front, one can suggest that you stop or turn, which is a simple rule. So our machinery could potentially be applied at a local level where the reference model is interpretable in that locality. This might help in "spot checking" a deployed model and estimating its safety by computing these maximum deviations in scrupulously selected (challenging) scenarios.

**Impossible inputs in certification set**   Mathematically simple sets such as Cartesian products and $\ell_p$ balls permit simpler algorithms for optimizing functions over them. Accordingly, these sets have been the focus of not only the present work but also the related literature on ML verification and adversarial robustness. However, they may not serve to exclude inputs that are physically or logically impossible from the certification set $\mathcal{C}$, and thus, the resulting maximum deviation values may be too large and conservative. Here it is important to distinguish between impossible inputs and those that are merely implausible (i.e., with low probability). Techniques for capturing implausibility have been proposed for contrastive/counterfactual explanations (Dhurandhar et al., 2018; Luss et al., 2021), whereas we expect the set of impossible inputs to be smaller and more constrained. As a simple example from the Adult Income dataset, if we agree that a wife/husband is defined to be of female/male gender (regardless of the gender of the spouse), then the cross combinations male-wife and female-husband cannot occur. Future work can consider the representation and handling of such constraints.

**White-box vs. grey-box models**   In this paper, we have assumed full "white-box" access to both $f$ and $f_0$, namely complete knowledge of their structure and parameters. Interesting questions may

arise when this assumption is relaxed to different "grey-box" possibilities. For example, one could further investigate the third case in Section 4.4, where one of $f, f_0$ is black-box and the other is white-box interpretable. There may exist assumptions that we have not identified that would improve the query complexity compared to generic black-box optimization.

**Other interpretability-safety relationships** This paper has focused on one relationship between the interpretability of a model and the safety of its outputs. It has not addressed other ways in which interpretability/explainability can affect the *risk* of a model (in the plain English sense, not the expectation of a loss function). For example, in regulated industries such as consumer finance, not providing explanations or providing inadequate ones can lead to legal, financial, and reputational risks. On the other hand, providing explanations is associated with its own risks (Weller, 2019). These include the leakage of personal information or model information (intellectual property), an increase in appeals of decisions for the decision-making entity, and strategic manipulation of attributes (i.e. "gaming") by individuals to gain more favorable outcomes.

**Applicability to RL settings** In RL, if one views the actions as labels and state representation as features, one can build a tree, albeit likely a deep/wide one, to represent exactly the RL policy, where the probability distribution over the actions can be viewed as the class distribution in a normal supervised setting. Rolling up the states, creating leaves with multiple states, and simply averaging the probabilities for each action would yield smaller trees that approximate the policy. Our work lays a foundation where in principle we can also compare $f$ and $f_0$ that are policies using such tree representations. This may be related to a popular global explainability method (Bastani et al., 2018) that samples policies and builds trees to explain them.

## D ADDITIONAL EXPERIMENTAL RESULTS

### D.1 ADULT INCOME DATASET

**Reference model** Figure 3 depicts the 8-leaf DT reference model used in the experiments on the Adult Income dataset. The root node separates individuals based on whether the marital status is *Married-civ-spouse*. The remaining splits divide the population into those with high and low education, high and low capital gains, and high and low capital losses. In particular, having high capital gains or losses is a good predictor of high income ($> \$50000$).

| $C$ | nonzeros | $\ell_1$ norm | accuracy | AUC |
|---|---|---|---|---|
| 3e-4 | 1 | 0.2 | 0.764 | 0.715 |
| 1e-3 | 6 | 2.6 | 0.825 | 0.885 |
| 3e-3 | 7 | 4.7 | 0.840 | 0.895 |
| 1e-2 | 16 | 7.4 | 0.848 | 0.900 |
| 3e-2 | 30 | 12.9 | 0.852 | 0.905 |
| 1e-1 | 38 | 17.3 | 0.853 | 0.905 |
| 3e-1 | 62 | 25.3 | 0.853 | 0.905 |
| 1e+0 | 83 | 40.7 | 0.852 | 0.905 |
| 3e+0 | 92 | 54.6 | 0.852 | 0.905 |
| 1e+1 | 101 | 65.1 | 0.852 | 0.904 |
| 3e+1 | 105 | 73.5 | 0.852 | 0.904 |
| 1e+2 | 107 | 77.6 | 0.852 | 0.904 |
| 3e+2 | 107 | 79.1 | 0.852 | 0.904 |

Table 1: Number of nonzero coefficients, $\ell_1$ norm of coefficients, test set accuracy, and area under the receiver operating characteristic (AUC) for logistic regression models on the Adult Income dataset as a function of inverse $\ell_1$ penalty $C$.

**LR and GAM models** In Tables 1 and 2, we show the values of inverse $\ell_1$ penalty $C$ and `max_bins` that were used for LR and GAM respectively, as well as statistics of the resulting classifiers. Test set accuracy and area under the receiver operating characteristic (AUC) increase and reach a plateau. For LR, we take the $\ell_1$ norm of the coefficients to be the main measure of smoothness as it depends on both the number of nonzero coefficients as well as their magnitudes, which both

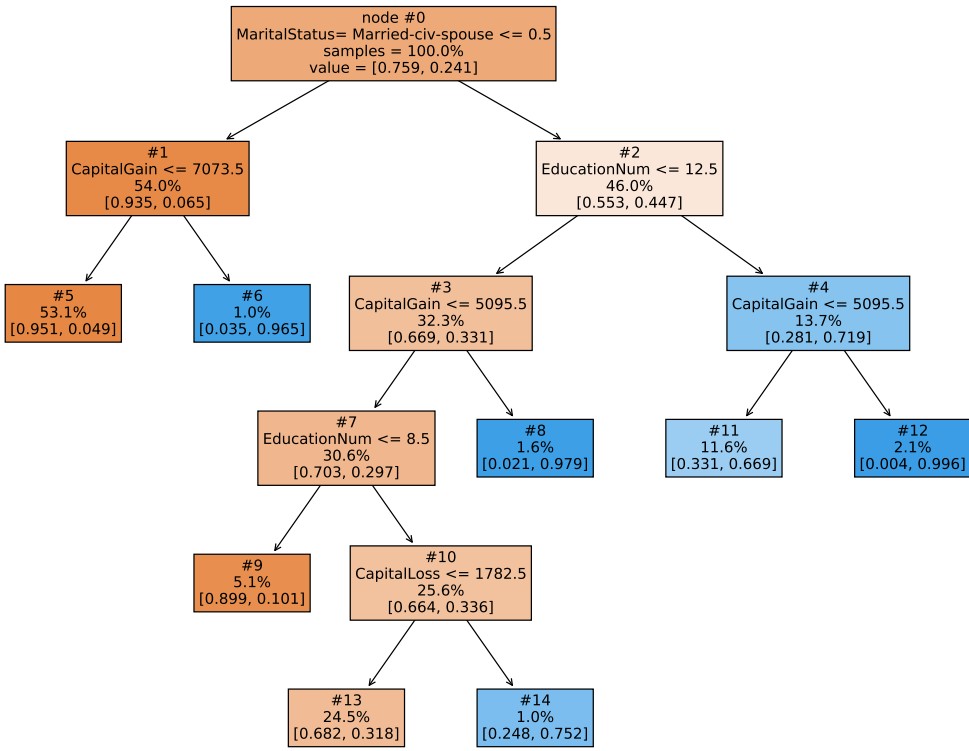

Figure 3: Decision tree reference model with 8 leaves for the Adult Income dataset.

| max_bins | accuracy | AUC |
|---|---|---|
| 4 | 0.858 | 0.910 |
| 8 | 0.862 | 0.915 |
| 16 | 0.865 | 0.920 |
| 32 | 0.870 | 0.924 |
| 64 | 0.871 | 0.925 |
| 128 | 0.871 | 0.925 |
| 256 | 0.872 | 0.925 |
| 512 | 0.871 | 0.925 |
| 1024 | 0.871 | 0.925 |

Table 2: Test set accuracy and AUC for Explainable Boosting Machines on the Adult Income dataset as a function of max_bins parameter.

affect the extreme values attained in (6). Based in part on Tables 1 and 2, we select $C = 0.01$ and max_bins $= 8$ as representative models that remain simple and have accuracies and AUCs not far from the maximum attainable. Plots for these two models are shown in Figures 4 and 5.

**Maximum deviation for decision trees with small numbers of leaves** In Figure 6, we expand upon the second panel in Figure 1 by focusing on decision trees with fewer than 50 leaves (plotted with linear spacing). The maximum deviation is 0.114 for trees with 9 and 10 leaves, and remains moderate up to 26 leaves.

**Relationships with accuracy and robust accuracy** In Figure 7, we show maximum deviation as a function of test set accuracy for the DT, LR, and GAM models shown in the right three panels of Figure 1 (the LR and GAM models are listed in Tables 1 and 2). Broadly, the plots show two

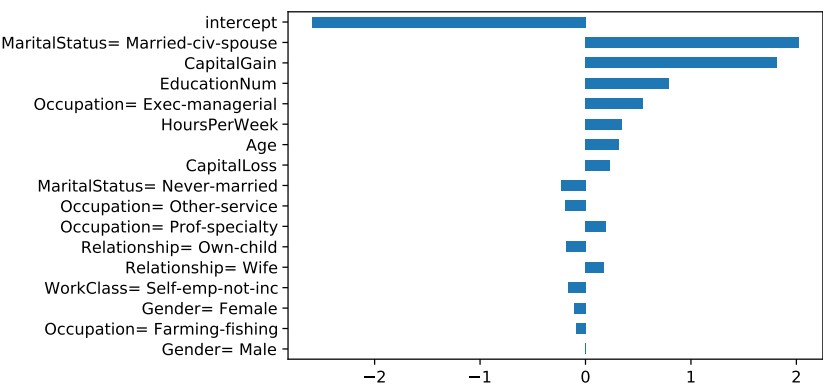

Figure 4: Coefficient values of the logistic regression model with $C = 0.01$ (16 nonzeros) for the Adult Income dataset.

regimes: one where accuracy increases and maximum deviation increases moderately or not at all, and one where accuracy stalls while maximum deviation increases. The latter is less desirable as it suggests increasing safety risks without a gain in accuracy. The last branch of the DT curve actually decreases in accuracy, indicating overfitting, while maximum deviation is high.

We also consider the relationship of maximum deviation to *robust accuracy*. Following Wong & Kolter (2018), robust loss for a pair $(x, y)$ is defined as the worst-case loss over an $\ell_\infty$ ball centered at $x$,

$$\max_{\|\Delta\|_\infty \leq \epsilon} L(f(x + \Delta), y), \tag{16}$$

and robust accuracy is therefore 1 minus the average robust 0-1 loss over a dataset. While Wong & Kolter (2018) focus on bounding robust loss for feedforward neural networks with ReLU activations, we find that the results in Section 4.2 apply to computing robust loss (16) exactly for LR and GAM models. Specifically, for 0-1 loss and $\ell_\infty$ balls, the separable optimization (8) applies, and the worst case is obtained by minimizing $f$ when the label $y$ is positive and maximizing $f$ when $y$ is negative.

The resulting robust accuracy values for DT, LR and GAM are plotted in Figure 8 in a similar fashion as Figure 7. Here we set $\epsilon = 0.1$ and $r = 0.1$ as well in computing maximum deviation. The DT plot shows maximum deviation increasing with model complexity while robust accuracy is stable up to a point. In the subsequent regime, when there are a large number of leaves, model robustness reduces while deviation remains high. The LR plot begins similarly to the one in Figure 7 in that robust accuracy increases along with maximum deviation, but then it stalls and decreases for maximum deviation above $0.94$. In the GAM plot, robust accuracy actually decreases with the `max_bins` parameter, i.e., the curve goes from right to left.

**Breakdown by leaves of $f_0$** In Figures 9–12, we plot the maximum log-odds achieved by model $f$ ($\max$ on RHS of (7)), the minimum log-odds achieved by $f$, and the reference model log-odds $g(y_{0m})$ over each leaf of the decision tree reference model in Figure 3. Plots are on the log-odds scale to show the deviations more clearly, including those that would be compressed by the nonlinear logistic function $g^{-1}(z) = 1/(1 + e^{-z})$. Figures 9 and 11 show the dependence on the certification set radius $r$ while Figures 10 and 12 show dependence on the smoothness parameters for LR and GAM. These figures provide a more granular picture corresponding to the summary in Figure 1 and support the trends seen there. In Figures 9 and 11, there are jumps at $r = 1$ because this is the smallest radius that permits the values of categorical features of test set points (the ball centers in (2)) to change to any other value. (Recall that categorical features are one-hot encoded into binary-valued features.) In Figure 11, the GAM achieves the limiting deviations corresponding to $r = \infty$ ($\mathcal{C} = \mathcal{X}$, dashed lines) no later than $r = 1.2$. In Figure 9, the LR model achieves the lower limit on log-odds as soon as $r > 1$ but the upper limit is not achieved for most leaves.

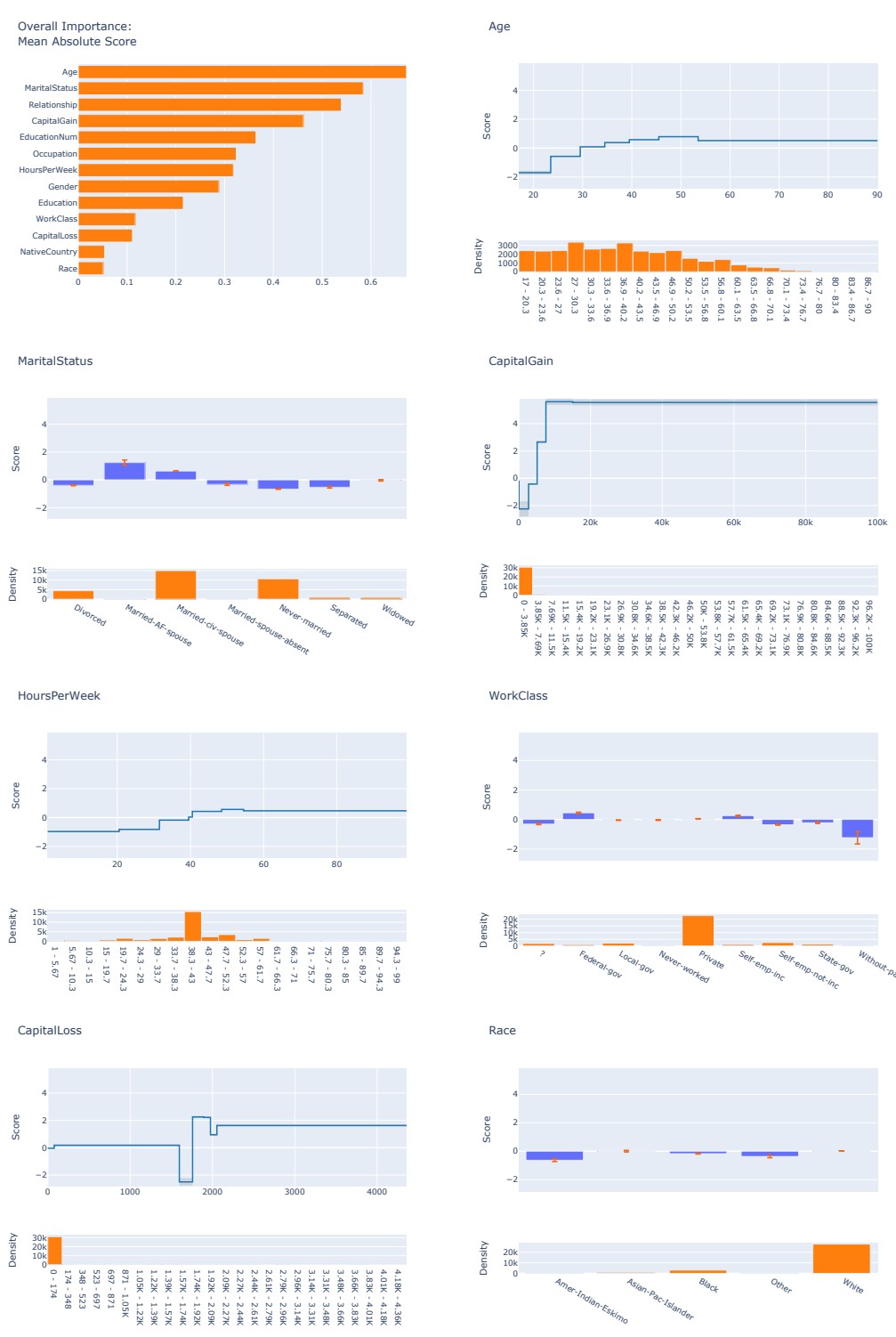

Figure 5: Feature importances and selected univariate functions $f_j$ for the Explainable Boosting Machine with max_bins = 8 on the Adult Income dataset.

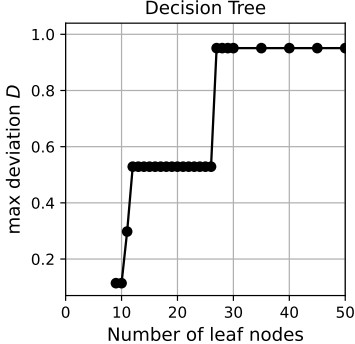

Figure 6: Maximum deviation $D$ for decision trees on the Adult Income dataset as a function of the number of leaves.

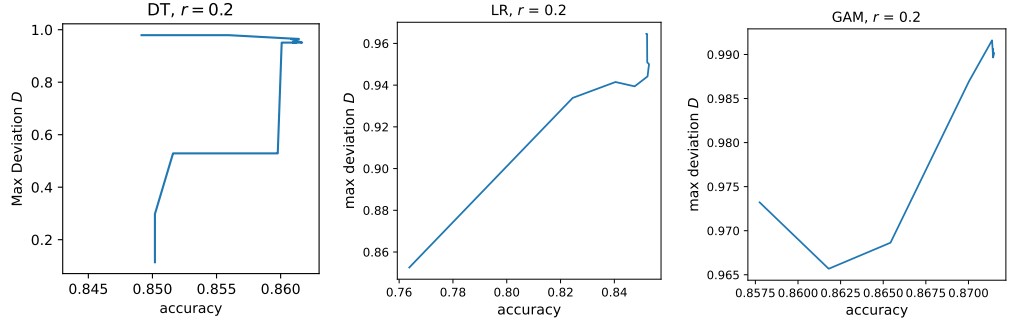

Figure 7: Maximum deviation $D$ (at certification set radius $r = 0.2$) vs. test set accuracy on the Adult Income dataset.

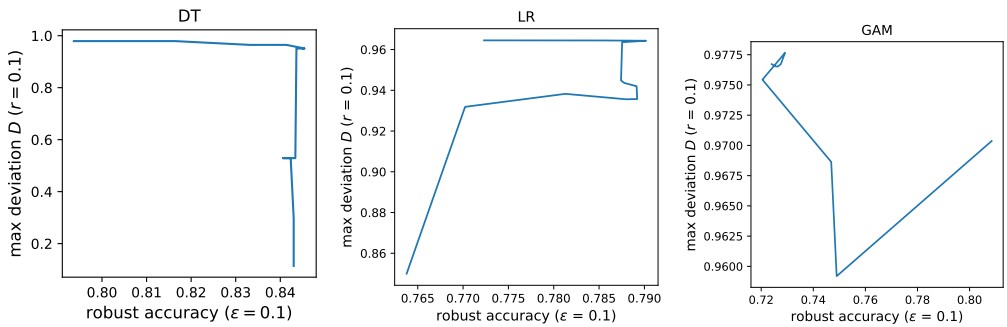

Figure 8: Maximum deviation $D$ vs. robust accuracy ($\epsilon = 0.1$, $r = 0.1$) on the Adult Income dataset.

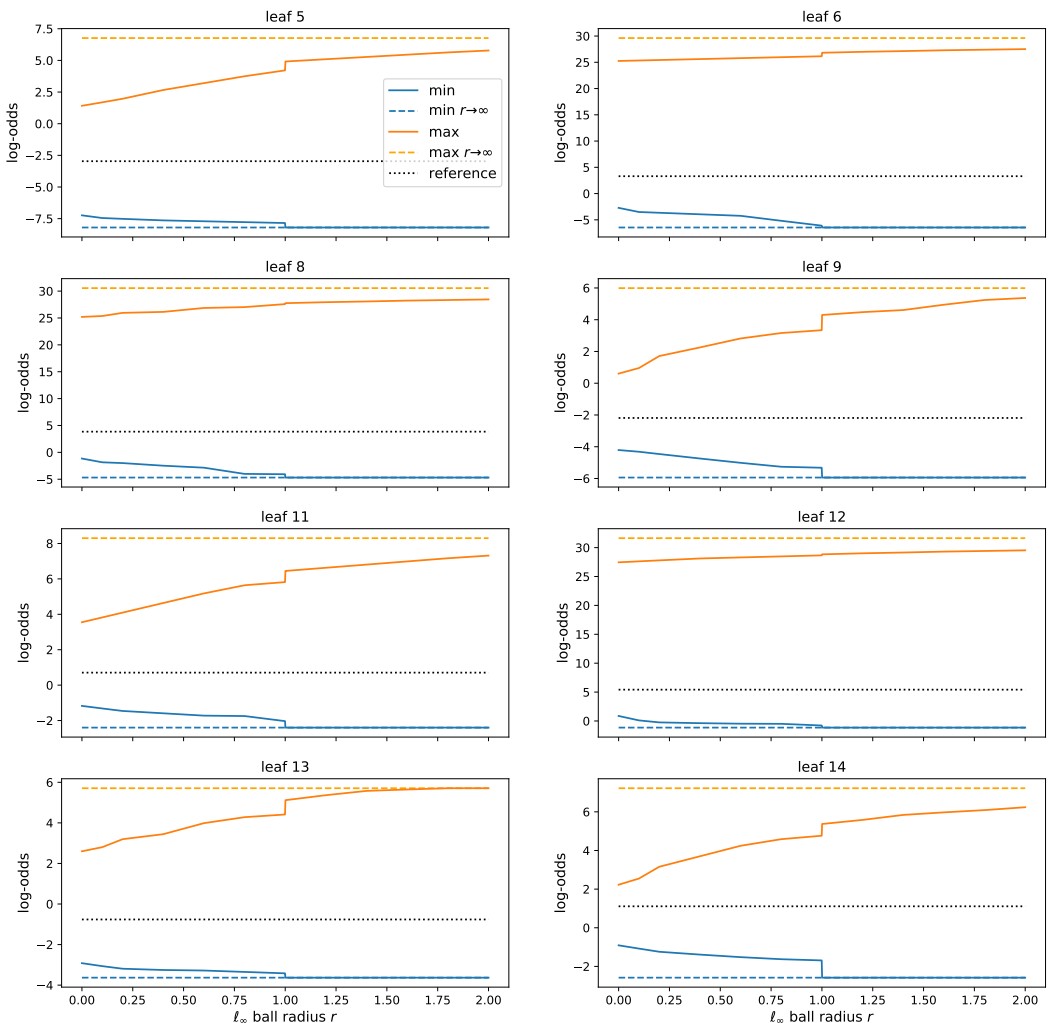

Figure 9: Minimum and maximum predicted log-odds for a logistic regression model with inverse $\ell_1$ penalty $C = 0.01$, as a function of certification set size (radius $r$) and broken down by leaves of the decision tree reference model.

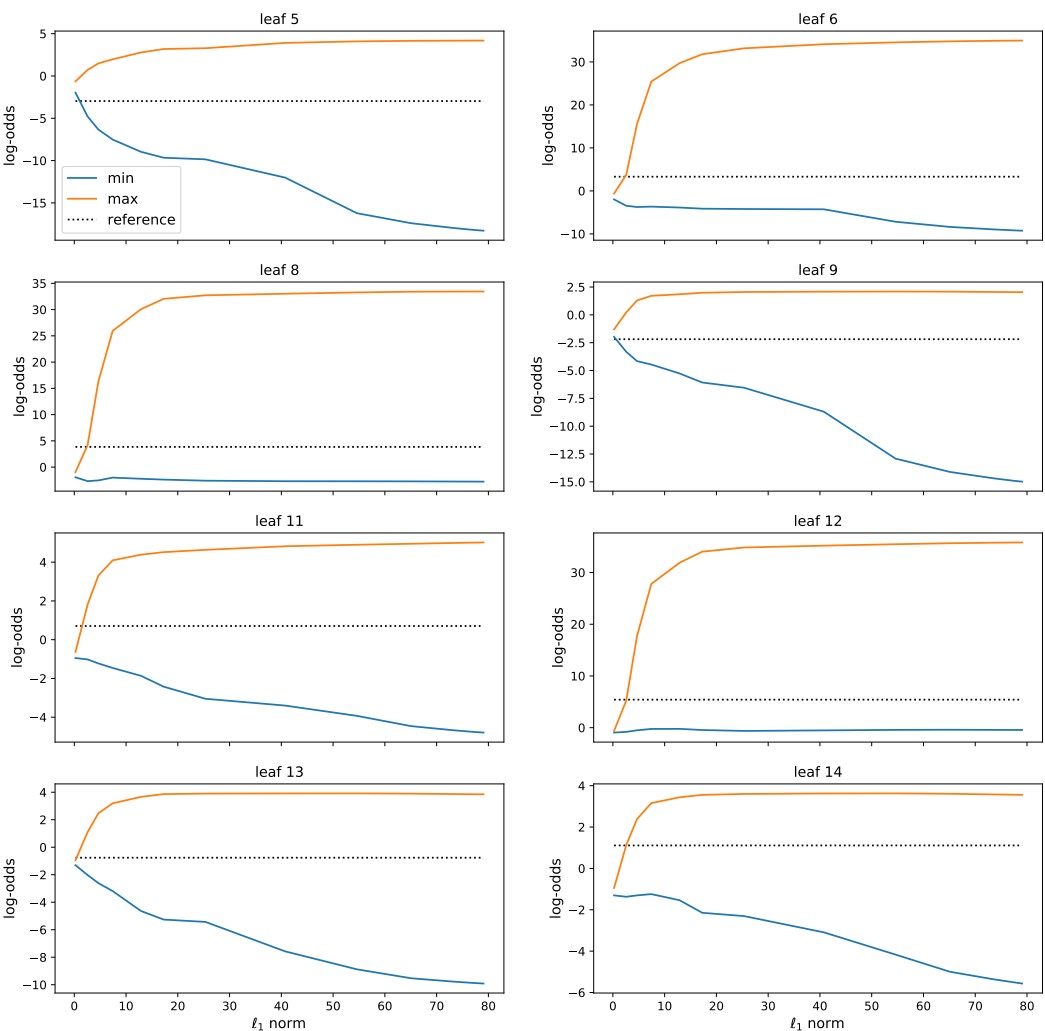

Figure 10: Minimum and maximum predicted log-odds for logistic regression models with different $\ell_1$ penalties $C$, broken down by leaves of the decision tree reference model. The certification set $\ell_\infty$ ball radius is $r = 0.2$.

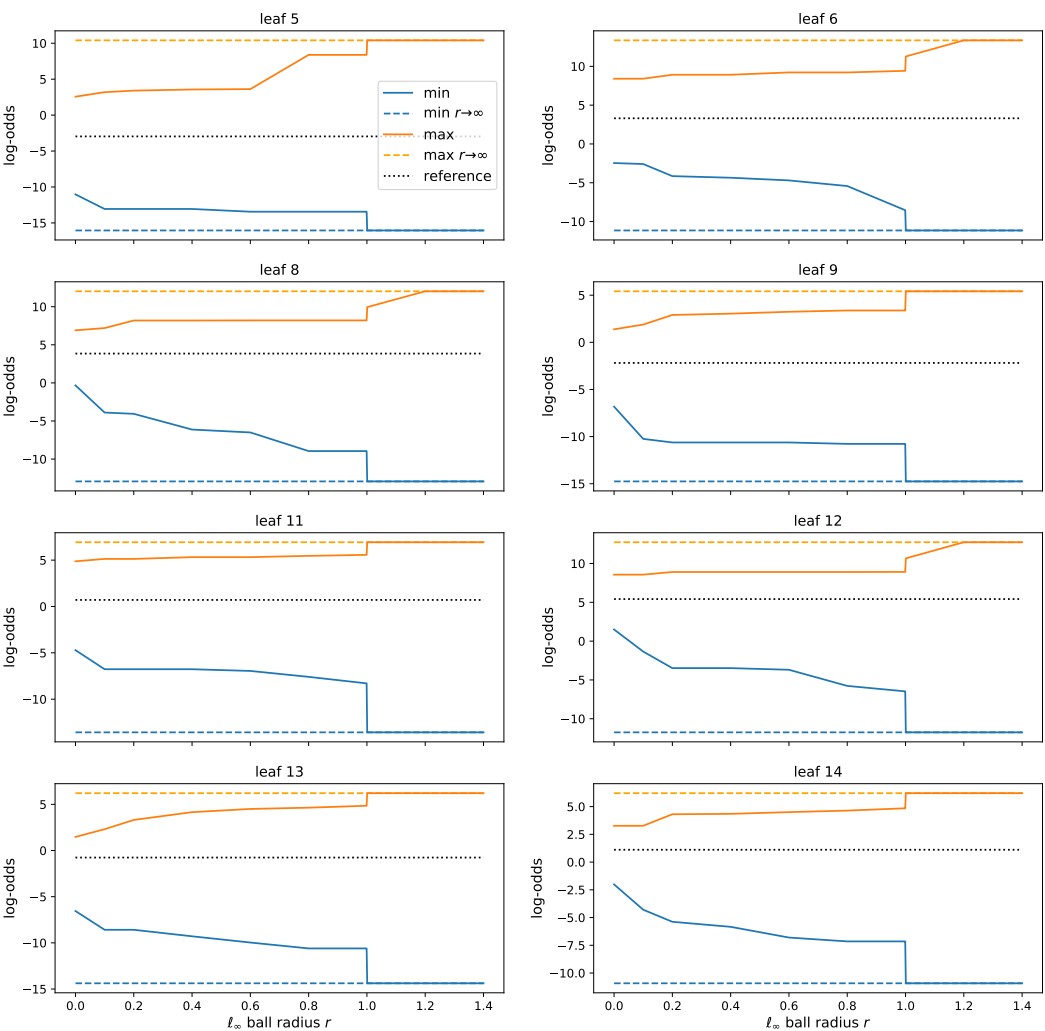

Figure 11: Minimum and maximum predicted log-odds for an Explainable Boosting Machine with max_bins = 8, as a function of certification set size (radius $r$) and broken down by leaves of the decision tree reference model.

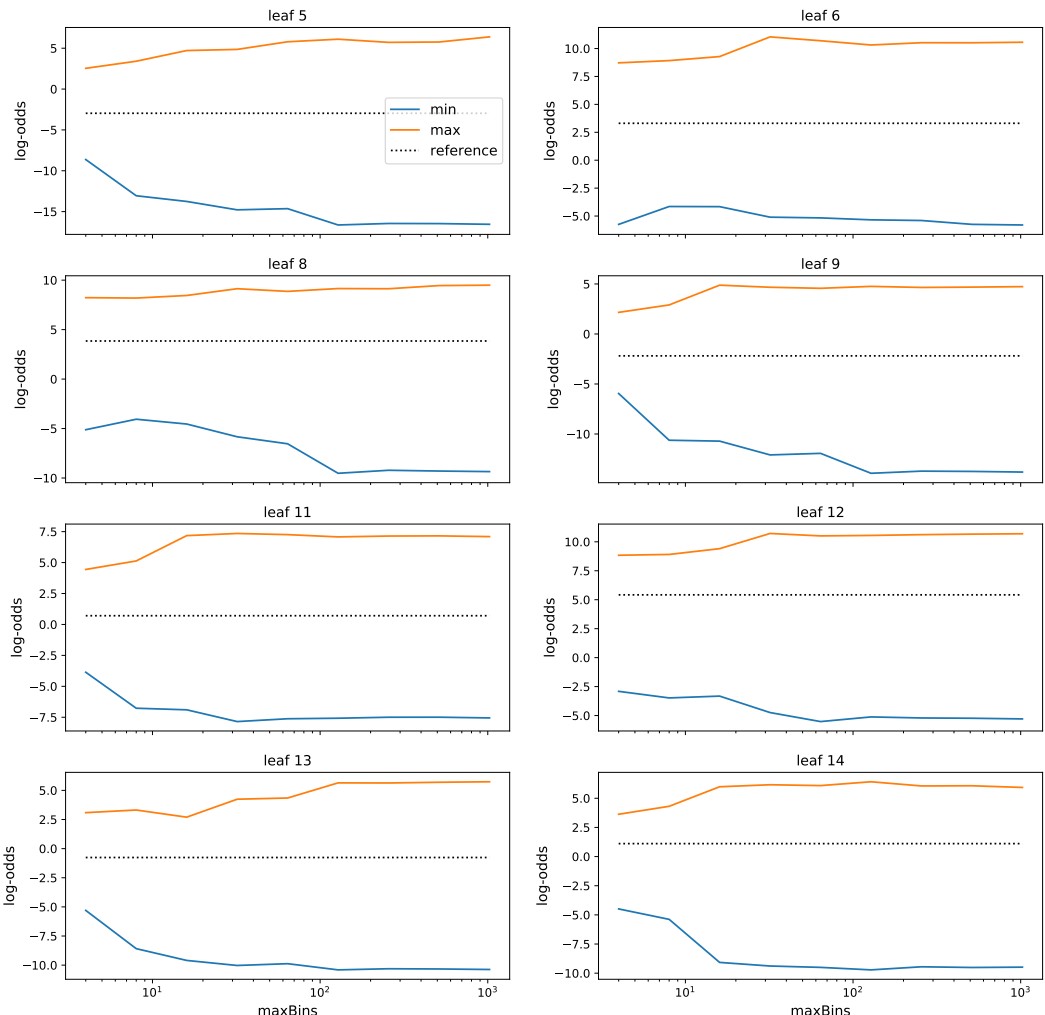

Figure 12: Minimum and maximum predicted log-odds for Explainable Boosting Machines with different max_bins values, broken down by leaves of the decision tree reference model. The certification set $\ell_\infty$ ball radius is $r = 0.2$.

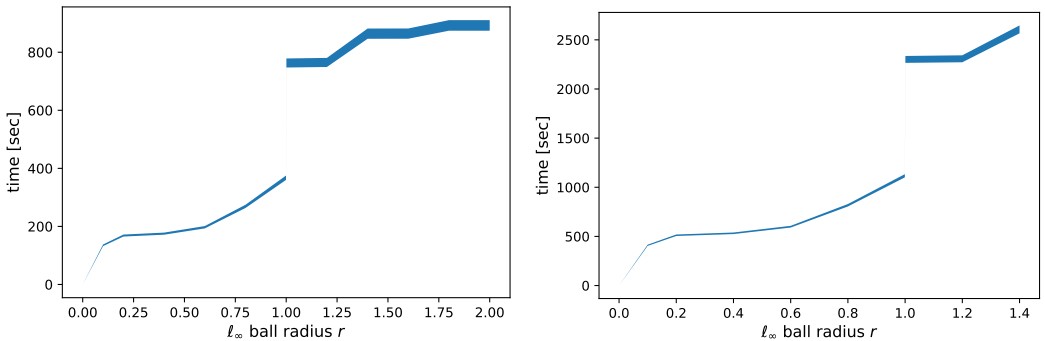

Figure 13: Time to compute maximum deviation for logistic regression models (left) and Explainable Boosting Machines (right) on the Adult Income dataset as a function of certification set size (radius $r$). The filled-in region shows the min-max variation with model complexity ($\ell_1$ norm for LR, `max_bins` for EBM).

**Running time** Figure 13 shows the time required to compute the maximum deviation for LR and GAM on the Adult Income dataset. These times were obtained using a single 2.0 GHz core of a server with 64 GB of memory (only a small fraction of which was used) running Ubuntu 16.04 (64-bit). The times increase with the $\ell_\infty$ ball radius $r$ because of the increasing number of ball-leaf intersections that become non-empty and hence need to be evaluated. The time for $r = 0$ is minimal because this case requires only model evaluation over the finite test set, as mentioned. The jumps at $r = 1$ are due again to the ability of categorical features to change values, leading to an increase in ball-leaf intersections. The filled-in regions show that there was little variation due to different $\ell_1$ norms for LR or `max_bins` for GAM. This was most likely because of a vectorized implementation, which operates on all LR coefficients or all GAM bins at once (i.e., without a for loop).

**Maximal cliques evaluated for DT, RF** To investigate the effectiveness of pruning by bounds in Algorithm 1, we investigate the number of times all the decision trees in the Random Forest have to be processed. This represents number of times the state could not be pruned and needed to be evaluated fully.

Figure 14 shows two aspects at play. (a) Pruning by bound is effective in restricting the search space more so for Random Forests than for decision trees, and (b) for larger graphs, more time is spent in computing bounds in Eq. (11).

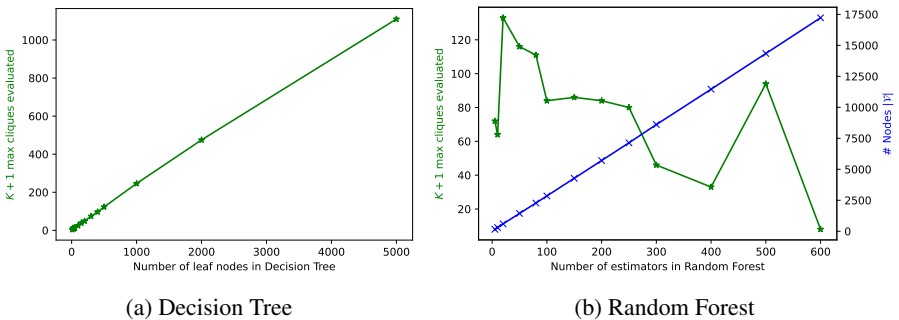

(a) Decision Tree            (b) Random Forest

Figure 14: Effectiveness of pruning by bound for tree-based models

**Feature combinations that maximize deviation** In Tables 3 and 4, we report feature values that maximize deviation over selected leaves of the DT reference model, for LR and GAM respectively. For Table 3, we have chosen the minimum log-odds over leaf 6 (corresponding to Figure 9, leaf 6, blue curve), which is one of the two leaves $m$ in (7) that maximize the deviation overall (the other being leaf 8). For Table 4, the minimum log-odds over leaf 12 is chosen (corresponding to Figure 11, leaf 12, blue curve) because this maximizes the deviation overall for most values of $r$.

The tables show the 6 features that contribute most to the minimum log-odds. These contributions are determined using (8) (with max replaced by min); since the minimum log-odds occurs in one of the $\ell_\infty$ ball-leaf intersections and this intersection is a Cartesian product, the decomposition in (8) applies. The contribution of feature $j$ is then $\min_{x_j \in \mathcal{S}_j} f_j(x_j)$. To account for the multiple $r$ values, we take an average of the contributions over $r$.

| $r$ | EducationNum | HoursPerWeek | Age | MaritalStatus | Occupation | Relationship |
|---|---|---|---|---|---|---|
| 0.0 | 9.0 | 15.0 | 23.0 | Never-married | Sales | Own-child |
| 0.1 | 4.7 | 33.8 | 26.6 | Never-married | Transport-moving | Not-in-family |
| 0.2 | 4.5 | 32.5 | 25.3 | Never-married | Transport-moving | Not-in-family |
| 0.4 | 4.0 | 30.1 | 22.5 | Never-married | Transport-moving | Not-in-family |
| 0.6 | 3.5 | 27.6 | 19.8 | Never-married | Transport-moving | Not-in-family |
| 0.8 | 1.0 | 26.1 | 17.0 | Never-married | Farming-fishing | Not-in-family |
| 0.999 | 1.0 | 7.7 | 17.0 | Never-married | Other-service | Own-child |
| 1.001 | 1.0 | 1.0 | 17.0 | Never-married | Other-service | Own-child |
| 1.2 | 1.0 | 1.0 | 17.0 | Never-married | Other-service | Own-child |
| 1.4 | 1.0 | 1.0 | 17.0 | Never-married | Other-service | Own-child |
| 1.6 | 1.0 | 1.0 | 17.0 | Never-married | Other-service | Own-child |
| 1.8 | 1.0 | 1.0 | 17.0 | Never-married | Other-service | Own-child |
| 2.0 | 1.0 | 1.0 | 17.0 | Never-married | Other-service | Own-child |
| $\infty$ | 1.0 | 1.0 | 17.0 | Never-married | Other-service | Own-child |

Table 3: Feature values that minimize log-odds for a logistic regression model ($C = 0.01$) over leaf 6 of the decision tree reference model. The 6 features that contribute most to the minimum are shown as a function of certification set radius $r$.

As $r$ increases, the predominant trend of the values of continuous features is toward extremes of the domain $\mathcal{X}$, depending on the sign of the corresponding LR coefficient $w_j$ or shape of the GAM function $f_j$. For example, EducationNum (education on an ordinal scale), hours per week, and age decrease toward minimum values, while capital gain occupies the minimal interval permitted for leaf 12 (see Figure 3). (These examples make sense since the log-odds of high income is being minimized.) This movement toward extremes is expected in the LR case because the functions $w_j x_j$ are either increasing or decreasing, and it is also true for GAM if the function $f_j$ is mainly increasing or decreasing. The values sometimes change abruptly in the opposite direction, for example hours per week in both Tables 3, 4, and age in the latter. These abrupt changes are due to the minimum jumping from one ball in (2) to another as $r$ increases, but the overall trend eventually prevails. For categorical features, the trend is toward values that minimize $f_j(x_j)$, e.g., *Never-married* marital status, *Without-pay* work class. While the contribution of each of these features to minimizing log-odds may be limited, together they do add up.

| $r$ | CapitalLoss | Age | HoursPerWeek | WorkClass | CapitalGain | Race |
|---|---|---|---|---|---|---|
| 0.0 | 0 | 29.0 | 40.0 | Private | 7298 | White |
| 0.1 | [ 0 40] | [53.6 56.4] | [18.8 20.5] | ? | [5095 5119] | White |
| 0.2 | [ 0 78] | [23.3 23.5] | [4.5 9.5] | State-gov | [5095 5119] | White |
| 0.4 | [ 0 78] | [20.5 23.5] | [ 2.1 11.9] | State-gov | [5095 5119] | White |
| 0.6 | [ 0 78] | [28.8 29.5] | [30.6 31.5] | Private | [5095 5119] | Asian-Pac-Islander |
| 0.8 | [1598 1759] | [20.1 23.5] | [30.1 31.5] | State-gov | [5095 5119] | White |
| 0.999 | [1598 1759] | [21.4 23.5] | [27.7 31.5] | Private | [5095 5119] | Amer-Indian-Eskimo |
| 1.001 | [1598 1759] | [17. 23.5] | [11.6 20.5] | Without-pay | [5095 5119] | Amer-Indian-Eskimo |
| 1.2 | [1598 1759] | [17. 23.5] | [ 9.2 20.5] | Without-pay | [5095 5119] | Amer-Indian-Eskimo |
| 1.4 | [1598 1759] | [17. 23.5] | [ 6.7 20.5] | Without-pay | [5095 5119] | Amer-Indian-Eskimo |
| $\infty$ | [1598 1759] | [17. 23.5] | [ 1. 20.5] | Without-pay | [5095 5119] | Amer-Indian-Eskimo |

Table 4: Feature values that minimize log-odds for an Explainable Boosting Machine (max_bins = 8) over leaf 12 of the decision tree reference model. The 6 features that contribute most to the minimum are shown as a function of certification set radius $r$. For $r > 0$, the minimizing values of continuous features form an interval because the corresponding functions $f_j$ are piecewise constant.

A notable exception to the trend toward extremes is capital loss in Table 4. For this feature, the values settle into the intermediate interval $[1598, 1759]$. The plot of the CapitalLoss function in

Figure 5 shows that this interval corresponds to a curiously low value of the function. This low region may be an artifact since it seems anomalous compared to the rest of the CapitalLoss function, and since individuals who report capital losses on their income tax returns to offset capital gains usually have high income (hence high log-odds).

Given the results in Tables 3 and 4, one question that arises is whether the feature combinations are indeed possible, if not the ones for $r \to \infty$, then at least for some finite value of $r$. For the top features shown in the two tables, while some combinations may appear improbable (for example, EducationNum $= 1$ and 1 hour per week), we submit that none appear *impossible*. However, if one considers features beyond the top 6, then some "impossible" combinations do occur (e.g., a female husband), although the contributions of these features to the minimum log-odds are much less. We touch upon this issue again in Appendix C.

The next question one might consider is the implication of these maximal deviations. From Figure 3, it is seen that leaf 6 classifies individuals with high capital gains as high income with high probability (0.965). Leaf 12 adds the attributes of married status and high education, and hence classifies as high income with even higher probability (0.996). At the same time, the feature values in Tables 3 and 4, which minimize log-odds for LR and GAM, also make sense according to basic domain knowledge. For example, few hours per week and young age are associated with lower income, as are *Without-pay* work class and *Amer-Indian-Eskimo* race in the United States. When these conflicting associations occur in combination and the combination does not appear impossible, the question may be which one prevails. Such a question might be resolvable by a domain expert. Alternatively, the disagreement between models $f$ and $f_0$ on the extreme examples in Tables 3, 4 may be reason to be cautious about using either of the models in these cases. This might lead to a way of combining the models or abstaining from prediction altogether. Lastly, the anomalously low region in the CapitalLoss function identified in Table 4 is a clear, concrete example where further investigation is warranted.

## D.2 LENDING CLUB DATASET

This dataset consists of 2.26 million rows with 14 features on loans. The target variable is whether a loan will be paid-off or defaulted on. Features describe the terms of the loan, e.g. duration, grade, purpose, etc. and borrower financial information such as credit history and income. For this case study, we consider a loan approval scenario using only information available at the time of application. In particular, we exclude the feature 'total_pymnt' (total payment over time on the loan), which becomes known at essentially the same time as the target variable. (When 'total_pymnt' is included as a feature, the prediction task becomes easy and accuracies in the high 90% range are possible.)

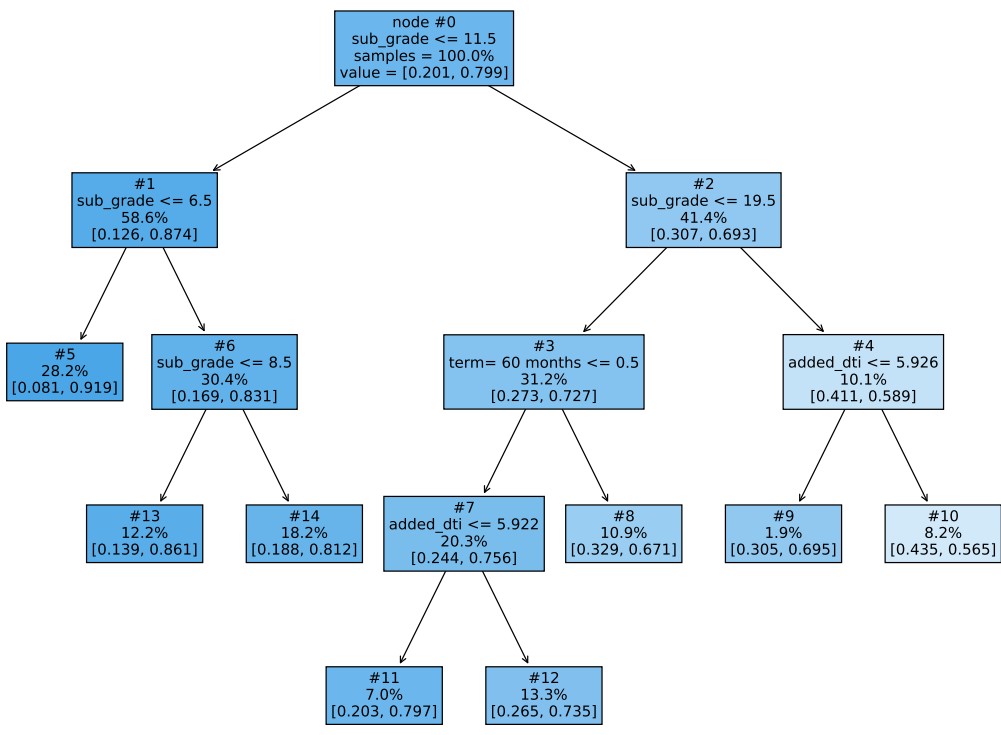

Figure 15: Decision tree reference model with 8 leaves for the Lending Club dataset.

**Reference model**    Figure 15 depicts the 8-leaf DT reference model for the Lending Club dataset. Most of the splits partition the 'sub_grade' feature, which is a measure of the quality of the loan (0–34 range, lower is better). Node 3 differentiates between 60-month terms and 36-month terms (the only alternative), while nodes 4 and 7 split on 'added_dti' (added debt-to-income ratio), which is the ratio between 12 months worth of the loan's payment installments and the borrower's annual income. While the structure of the DT agrees with domain knowledge (lower 'sub_grade' and lower 'added_dti' correlate with higher repayment probability), the test set accuracy of 79.8% is no better than that of the trivial predictor that always returns the majority class of "paid off". The DT's AUC of 0.689 however does indicate an improvement over the trivial predictor.

**LR and GAM models**    Tables 5 and 6 show the statistics of the LR and GAM classifiers that were trained on the Lending Club data. Similar to the DT reference model, the difference compared to Tables 1, 2 for the Adult Income dataset is that the accuracies remain no better than that of the trivial predictor, while the AUC does not show much increase either. These statistics suggest that the prediction task is difficult with the features available. Figures 16 and 17 display plots for the LR model with $C = 0.01$ and GAM with max_bins $= 8$, which are again chosen as representative

| $C$ | nonzeros | $\ell_1$ norm | accuracy | AUC |
|---|---|---|---|---|
| 1e-4 | 1 | 0.4 | 0.798 | 0.693 |
| 3e-4 | 2 | 0.6 | 0.799 | 0.695 |
| 1e-3 | 6 | 1.0 | 0.799 | 0.702 |
| 3e-3 | 8 | 1.3 | 0.799 | 0.702 |
| 1e-2 | 12 | 1.6 | 0.800 | 0.702 |
| 3e-2 | 17 | 2.1 | 0.799 | 0.703 |
| 1e-1 | 22 | 3.1 | 0.799 | 0.703 |
| 3e-1 | 24 | 3.7 | 0.799 | 0.703 |
| 1e+0 | 26 | 4.1 | 0.799 | 0.703 |
| 3e+0 | 26 | 4.2 | 0.799 | 0.703 |

Table 5: Number of nonzero coefficients, $\ell_1$ norm of coefficients, test set accuracy, and AUC for logistic regression models on the Lending Club dataset as a function of inverse $\ell_1$ penalty $C$.

| max_bins | accuracy | AUC |
|---|---|---|
| 4 | 0.798 | 0.696 |
| 8 | 0.798 | 0.703 |
| 16 | 0.799 | 0.704 |
| 32 | 0.799 | 0.705 |
| 64 | 0.799 | 0.705 |
| 128 | 0.799 | 0.705 |
| 256 | 0.799 | 0.705 |
| 512 | 0.799 | 0.705 |
| 1024 | 0.799 | 0.705 |

Table 6: Test set accuracy and AUC for Explainable Boosting Machines on the Lending Club dataset as a function of max_bins parameter.

models. The GAM in particular shows sensible monotonic behavior as functions of 'sub_grade', 'int_rate' (interest rate), 'dti' (debt-to-income ratio), etc., despite the unimpressive accuracy.

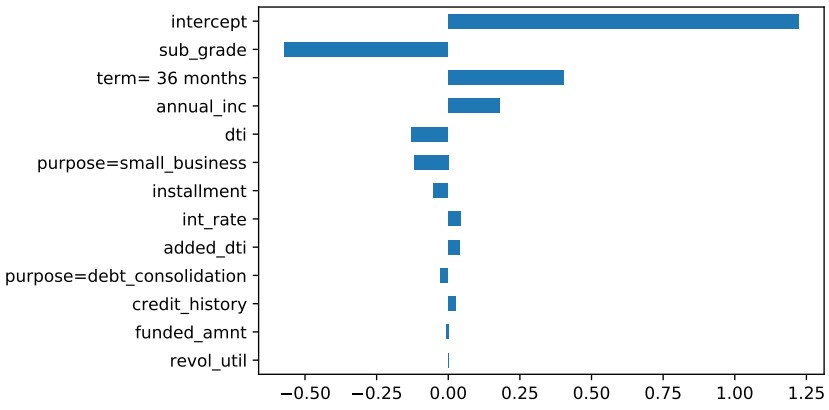

Figure 16: Coefficient values of the logistic regression model with $C = 0.01$ (12 nonzeros) for the Lending Club dataset.

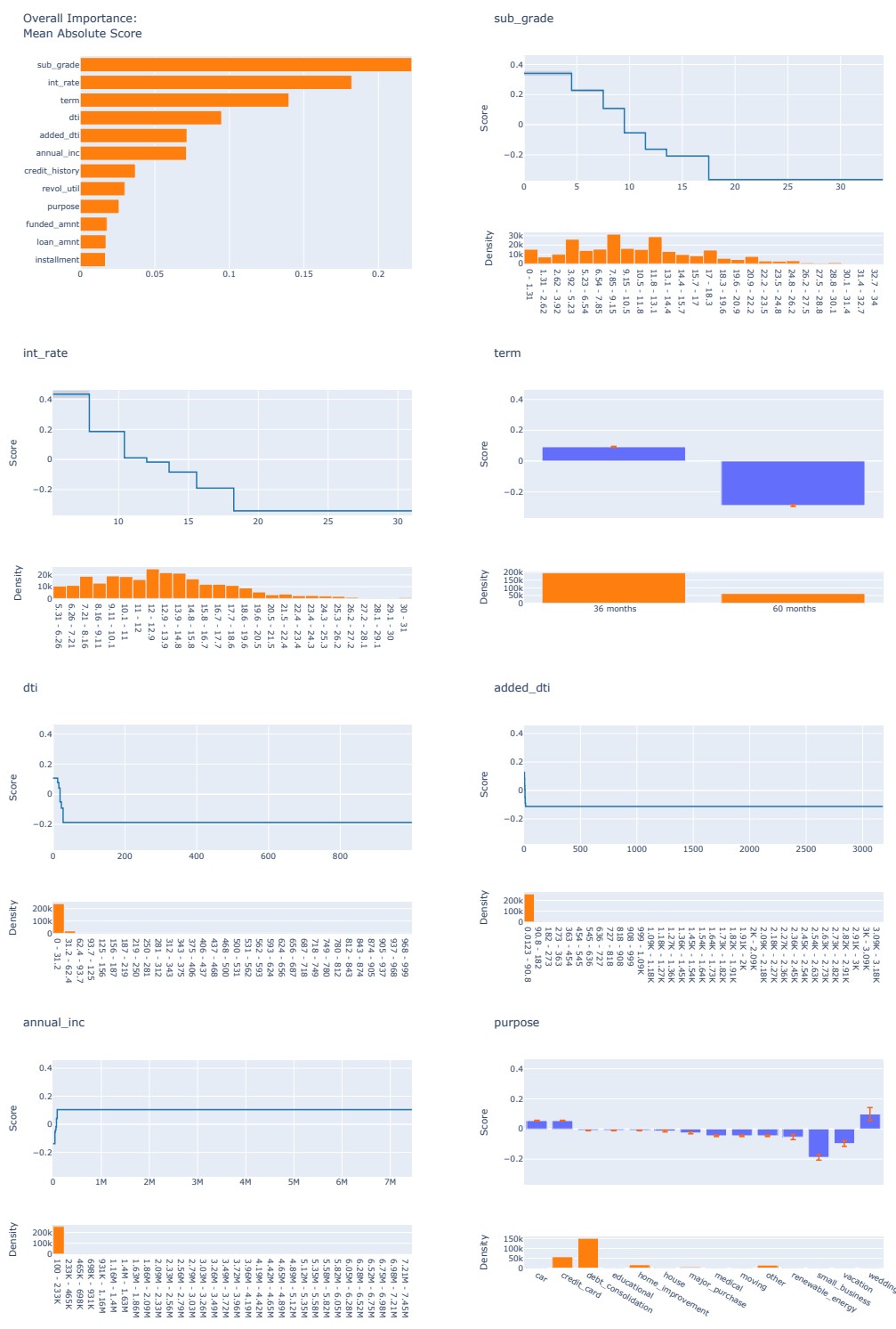

Figure 17: Feature importances and selected univariate functions $f_j$ for the Explainable Boosting Machine with max_bins = 8 on the Lending Club dataset.

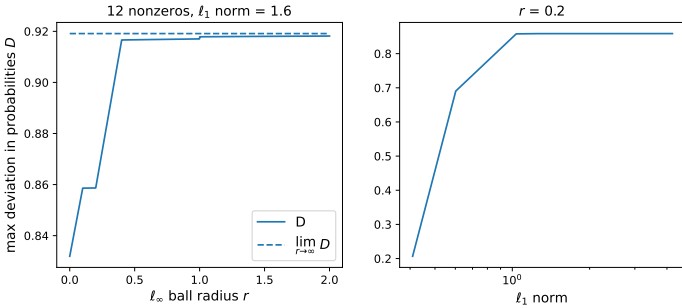

Figure 18: Maximum deviation $D$ for logistic regression models on the Lending Club dataset as a function of certification set size (radius $r$) and model smoothness ($\ell_1$ norm).

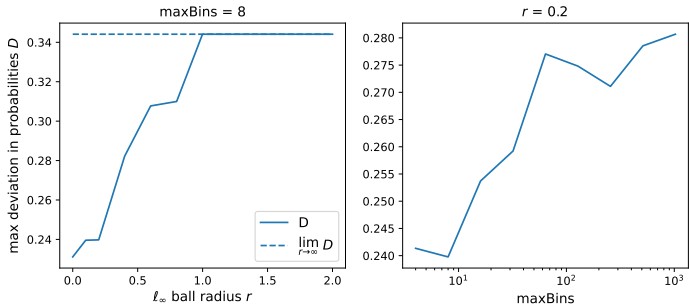

Figure 19: Maximum deviation $D$ for Explainable Boosting Machines on the Lending Club dataset as a function of certification set size (radius $r$) and model smoothness (`max_bins` parameter).

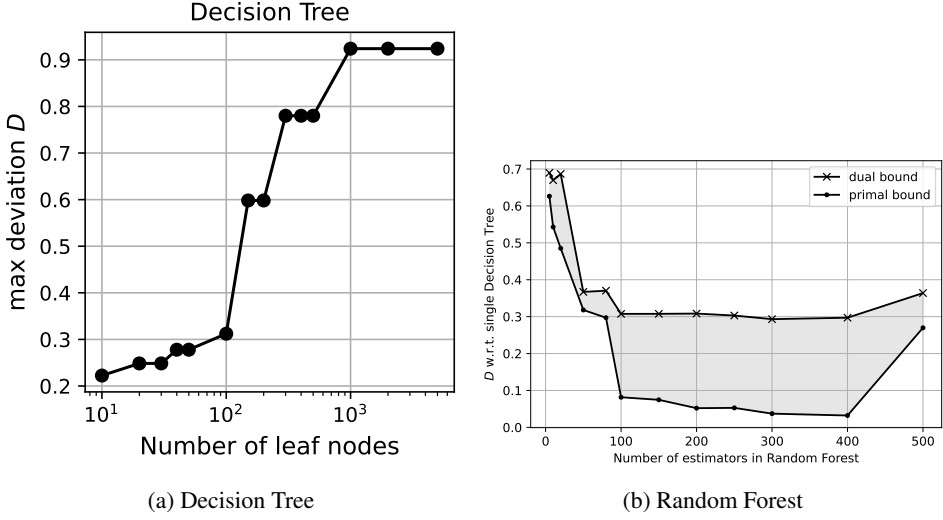

(a) Decision Tree

(b) Random Forest

Figure 20: Maximum deviations computed for tree and tree ensembles on the Lending Club dataset

**Maximum deviation summary** Figures 18–20 show maximum deviation as functions of certification set radius $r$ and model complexity parameters, in a similar manner as Figures 1 and 2 for the Adult Income dataset. The qualitative patterns are similar to before: increasing maximum deviation in all cases except with the number of RF estimators in Figure 20b, where the upper bound is stable around 0.7. A major quantitative difference is that the maximum deviations for the GAM in Figure 19 are much lower than for the other models, in particular LR in Figure 18. This is likely due to

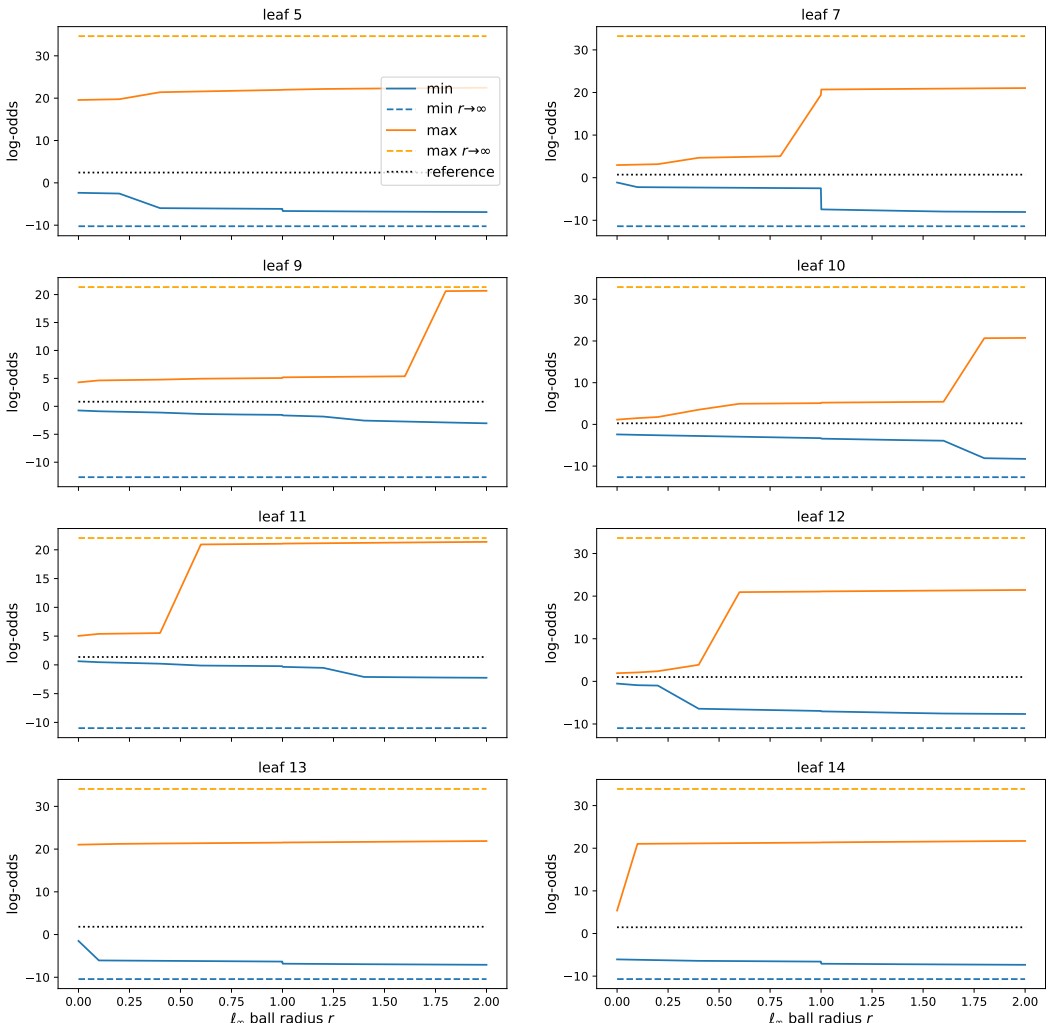

Figure 21: Minimum and maximum predicted log-odds for a logistic regression model with inverse $\ell_1$ penalty $C = 0.01$ on the Lending Club dataset, as a function of certification set size (radius $r$) and broken down by leaves of the decision tree reference model.

the fact that the GAM functions $f_j$ in Figure 17 are bounded while still being monotonic, unlike the linear functions $w_j x_j$ in the LR model.

**Breakdown by leaves of $f_0$**    Figures 21–24 show a breakdown of the deviations for LR and GAM by leaves of the reference model, similar to Figures 9–12 and again on the log-odds scale. One difference is that in Figure 21, the deviations for finite $r$ do not come close to their $r \to \infty$ counterparts in most cases. In Figure 23 however, the $r \to \infty$ values are all attained when $r$ is slightly greater than 1.

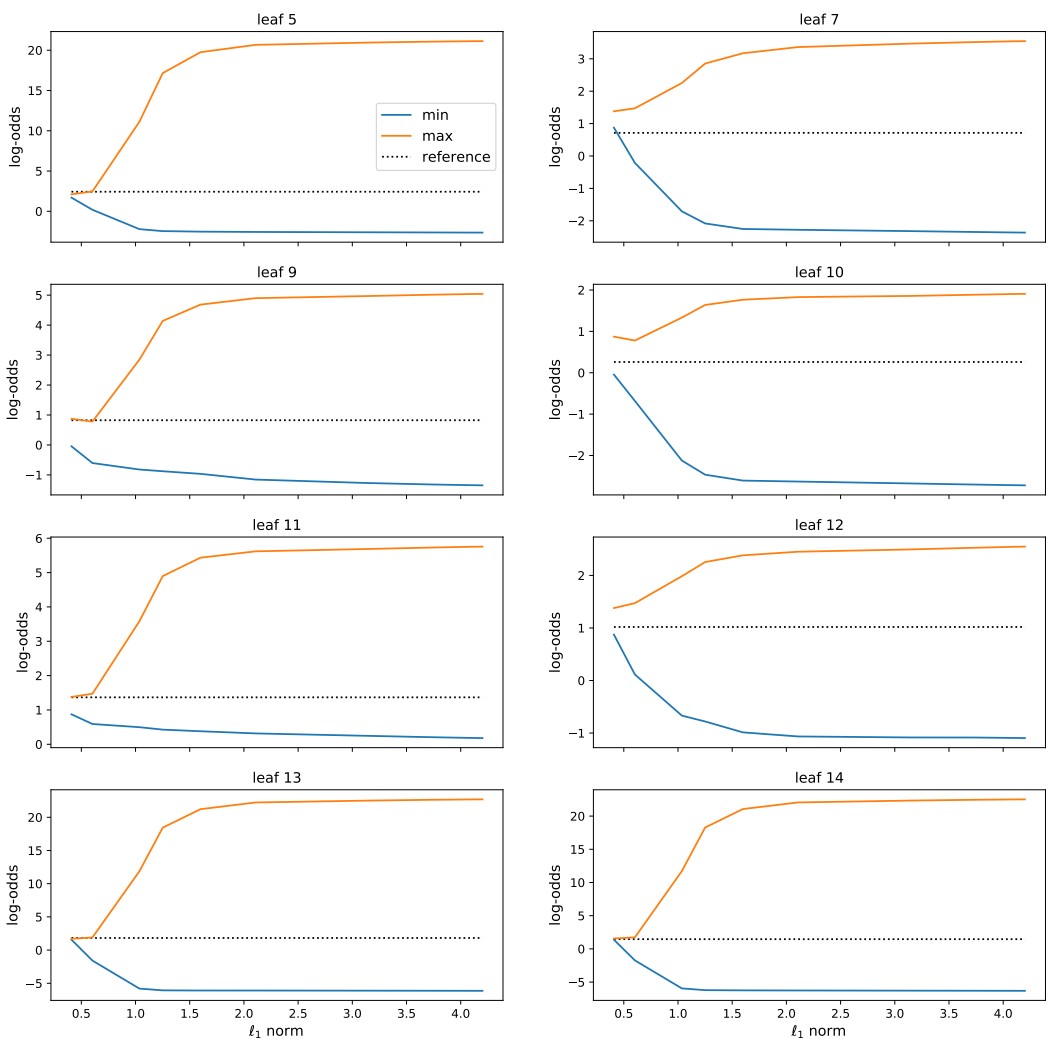

Figure 22: Minimum and maximum predicted log-odds for logistic regression models with different $\ell_1$ penalties $C$ on the Lending Club dataset, broken down by leaves of the decision tree reference model. The certification set $\ell_\infty$ ball radius is $r = 0.2$.

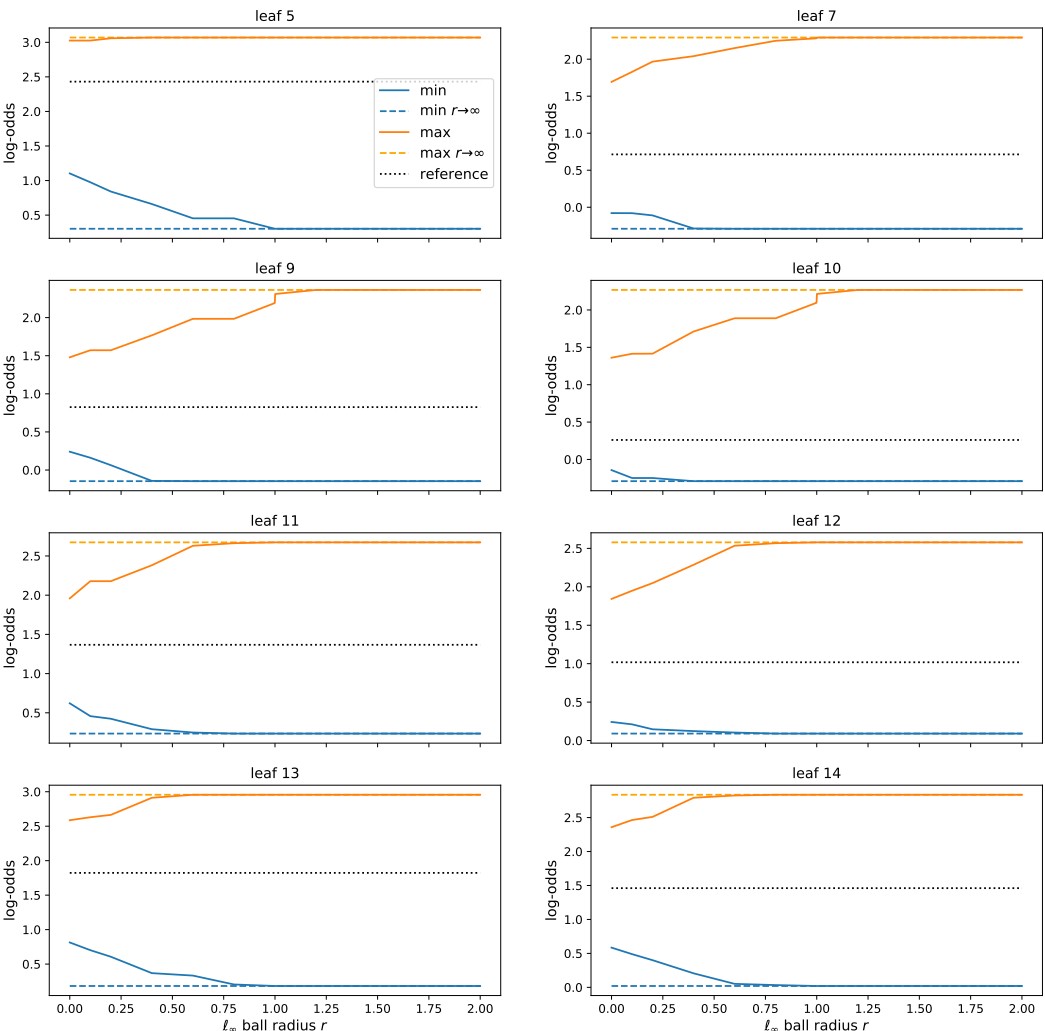

Figure 23: Minimum and maximum predicted log-odds for an Explainable Boosting Machine with max_bins = 8 on the Lending Club dataset, as a function of certification set size (radius $r$) and broken down by leaves of the decision tree reference model.

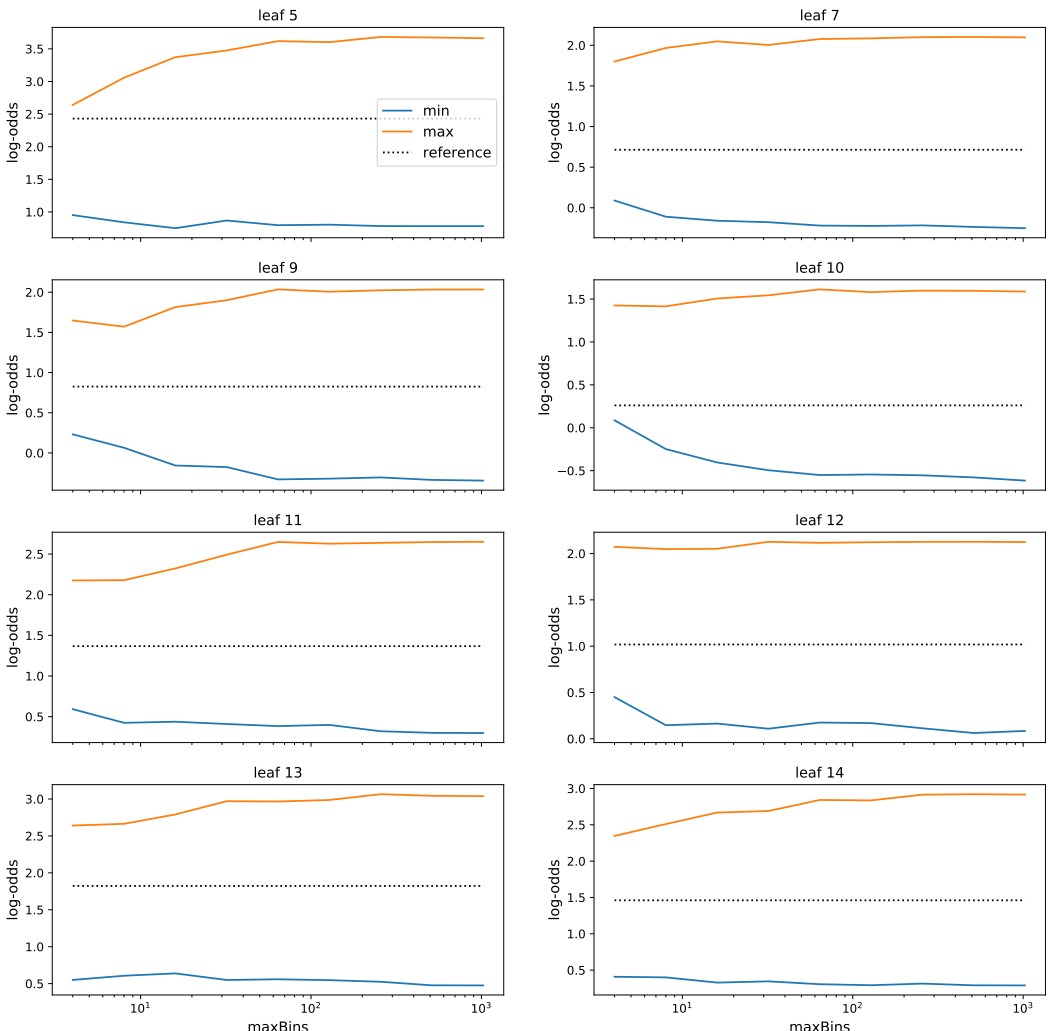

Figure 24: Minimum and maximum predicted log-odds for Explainable Boosting Machines with different max_bins values on the Lending Club dataset, broken down by leaves of the decision tree reference model. The certification set $\ell_\infty$ ball radius is $r = 0.2$.

**Running time, maximal cliques evaluated**   Figure 25 shows the time required to compute the maximum deviation for LR and GAM on the Lending Club dataset. Figure 26 shows the number of $K + 1$-maximal cliques evaluated for DT and RF as well as the number of nodes in the graph. The observations are the same as in Figures 13 and 14.

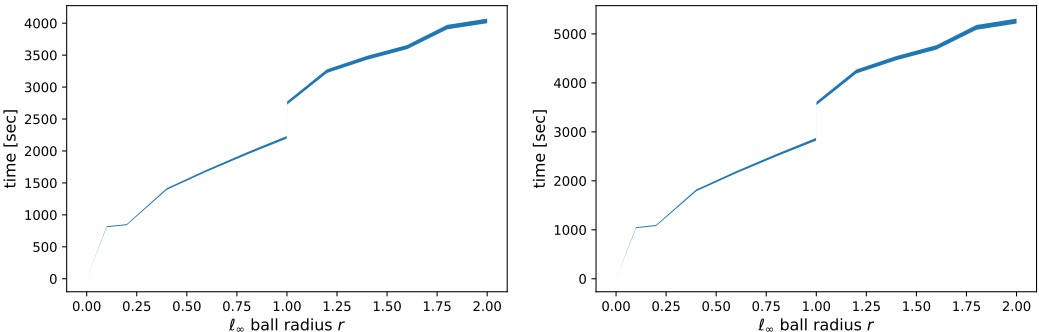

Figure 25: Time to compute maximum deviation for logistic regression models (left) and Explainable Boosting Machines (right) on the Lending Club dataset as a function of certification set size (radius $r$). The filled-in region shows the min-max variation with model complexity ($\ell_1$ norm for LR, `max_bins` for EBM).

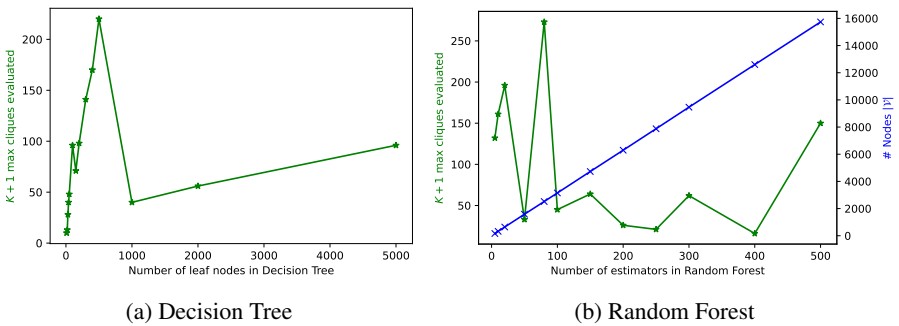

(a) Decision Tree                          (b) Random Forest

Figure 26: Effectiveness of pruning by bound for tree-based models on the Lending Club dataset

**Feature combinations that maximize deviation**   Tables 7 and 8 present the feature values that maximize deviation for LR and GAM respectively. For both tables, the minimum log-odds over leaf 5 of the DT reference model is selected (corresponding to Figures 21 and 23, leaf 5, blue curve) because this choice maximizes the deviation overall for most values of $r$.

The previous trend toward extreme feature values that minimize log-odds also holds here as $r$ increases. For example, debt-to-income ratios increase to outlier values above $100\%$, annual income drops to the minimum of $\$100$, the term changes to 60 months in Table 7 for $r > 1$, and the purpose changes to small business, the category with the lowest log-odds in Figure 17. The decrease in income and increases in debt-to-income ratios are qualitatively in accordance with each other. However, these quantities are related to each other by deterministic formulas (at least in theory) that also involve the interest rate and installment amount. It is not clear whether the values in Tables 7 and 8 violate these relationships. This may be an instance that could benefit from constraints on possible feature combinations, as briefly mentioned in Appendix C.

| $r$ | dti | annual_inc | term | purpose | int_rate | credit_history |
|---|---|---|---|---|---|---|
| 0.0 | 505 | 1700 | 36 months | credit_card | 7.4 | 2557 |
| 0.1 | 506 | 100 | 36 months | credit_card | 6.9 | 2283 |
| 0.2 | 507 | 100 | 36 months | credit_card | 6.4 | 2009 |
| 0.4 | 884 | 100 | 36 months | debt_consolidation | 10.1 | 3897 |
| 0.6 | 886 | 100 | 36 months | debt_consolidation | 9.1 | 3349 |
| 0.8 | 889 | 100 | 36 months | debt_consolidation | 8.2 | 2801 |
| 0.999 | 891 | 100 | 36 months | debt_consolidation | 7.2 | 2256 |
| 1.001 | 891 | 100 | 60 months | small_business | 7.2 | 2251 |
| 1.2 | 893 | 100 | 60 months | small_business | 6.3 | 1706 |
| 1.4 | 895 | 100 | 60 months | small_business | 5.3 | 1158 |
| 1.6 | 898 | 100 | 60 months | small_business | 5.3 | 1095 |
| 1.8 | 900 | 100 | 60 months | small_business | 5.3 | 1095 |
| 2.0 | 902 | 100 | 60 months | small_business | 5.3 | 1095 |
| $\infty$ | 999 | 100 | 60 months | small_business | 5.3 | 1095 |

Table 7: Feature values that minimize log-odds for a logistic regression model ($C = 0.01$) over leaf 5 of the decision tree reference model for the Lending Club dataset. The 6 features that contribute most to the minimum are shown as a function of certification set radius $r$.

| $r$ | term | int_rate | dti | purpose | annual_inc | added_dti |
|---|---|---|---|---|---|---|
| 0.0 | 60 months | 9.9 | 46.4 | debt_consolidation | 35000 | 25.5 |
| 0.1 | 60 months | [10.4 11. ] | [28.6 30.9] | debt_consolidation | [31800 38001] | [10. 11.2] |
| 0.2 | 60 months | [12. 13.1] | [27.8 31.8] | debt_consolidation | [36600 38001] | [10. 11.7] |
| 0.4 | 60 months | [13.6 14.3] | [27.8 30.4] | small_business | [ 100 38001] | [12.7 15.3] |
| 0.6 | 60 months | [15.6 16.3] | [27.8 36.1] | small_business | [19801 38001] | [12.7 15. ] |
| 0.8 | 60 months | [15.6 16.4] | [27.8 28.4] | small_business | [14402 38001] | [12.7 15.7] |
| 0.999 | 60 months | [18.2 18.4] | [27.8 38.8] | small_business | [33069 38001] | [12.7 14.5] |
| 1.001 | 60 months | [18.2 18.8] | [27.8 35.3] | small_business | [ 935 38001] | [12.7 18.5] |
| 1.2 | 60 months | [18.2 20.2] | [27.8 33.3] | small_business | [ 7602 38001] | [12.7 18.1] |
| 1.4 | 60 months | [18.2 21.2] | [27.8 35.6] | small_business | [ 100 38001] | [12.7 20.4] |
| 1.6 | 60 months | [18.2 19.2] | [27.8 29.9] | small_business | [ 100 38001] | [12.7 24.5] |
| 1.8 | 60 months | [18.2 26.1] | [27.8 28.5] | small_business | [ 100 38001] | [12.7 24.4] |
| 2.0 | 60 months | [18.2 27.1] | [27.8 30.7] | small_business | [ 100 38001] | [12.7 26.6] |
| $\infty$ | 60 months | [18.2 31. ] | [ 27.8 999. ] | small_business | [ 100 38001] | [ 12.7 3179.3] |

Table 8: Feature values that minimize log-odds for an Explainable Boosting Machine (`max_bins = 8`) over leaf 5 of the decision tree reference model for the Lending Club dataset. The 6 features that contribute most to the minimum are shown as a function of certification set radius $r$. For $r > 0$, the minimizing values of continuous features form an interval because the corresponding functions $f_j$ are piecewise constant.

