# OpenReview forum: "On the Safety of Interpretable Machine Learning: A Maximum Deviation Approach"
_ICLR.cc/2022/Conference — ICLR 2022 Submitted_

### Official Review · Reviewer_6m9J · 2021-10-22

**Correctness:** 4
**Technical Novelty And Significance:** 3
**Empirical Novelty And Significance:** 2
**Recommendation:** 5
**Confidence:** 4

**Main Review:**

Strong points of the paper include the following:

+ the main idea of the paper, i.e. using the maximum deviation to assess the safety of a model w.r.t. a baseline model, is well explained
+ the authors show that for some simple models, e.g. trees and rule lists, their proposed measure can be computed exactly
+ the authors propose a way to characterize interpretable functions via piecewise Lipschitz functions and give an interpretation in terms of tree models
+ the contributions seem to be technically correct
+ the approach is empirically validated and details to parameters and data sets are provided

Weaknesses include:
- When discussing related work, the authors mention that as opposed to previous work of Mohseni et al. (2021), their proposed approach can quantify safety through interpretability.  This is a quite bold statement, since the approach needs a reference model f_0 and judges safety of a new model f based on the difference compared to that model (f_0 needs to be provided). Then the question is, whether the baseline model is safe, which is difficult to answer by itself (especially if the baseline model is not very simple).
- Also, to me the proposed maximum deviation measure seems a bit conservative for evaluating the safety risk of a model. A large deviation between the prediction of two models could also mean that the new model is an improved estimator---i.e. it can model parts of the data that the old model did not model (put into the same leaf, for the example of decision trees).
- Last, it would be nice if the obtained score provides a guidance for domain experts to retrain a safer model in terms of the score.

Minor remark: Using H(S) in Eq. 12 is a bit confusing since H usually corresponds to entropy.


**Summary Of The Paper:**

The authors posit that the demand for explainable/interpretable models in machine learning is linked to safety. In other words, domain experts trust explainable or interpretable models more. Based on this premise, they suggest to assess the safety of a model by measuring its maximum deviation of a model to a reference model, which is supposed to be safe. This deviation is computed on a certification set of data points. Additionally, the authors show that their proposed maximum derivation can be computed exactly for some interpretable models, such as decision trees, rule lists and generalized linear and additive models. Further, they discuss implications for tree ensembles and empirically evaluate their approach.

**Summary Of The Review:**

In summary, the technical contributions and the evaluation of the approach seem solid. However, I do not see how exactly the proposed dissimilarity (the maximum deviation) can help a domain expert to retrain a safer model based in its output, which seems like the goal of such a metric. Inspecting every subset of data points on which the new model and the baseline differ substantially might take a lot of time.

---

> ### Author Response · Authors · 2021-11-16
> **Response to Reviewer 6m9J**
>
> **Weaknesses:**
>
> > the authors mention that as opposed to previous work of Mohseni et al. (2021), their proposed approach can quantify safety through interpretability. This is a quite bold statement, since the approach needs a reference model f_0 and judges safety of a new model f based on the difference compared to that model (f_0 needs to be provided). Then the question is, whether the baseline model is safe, which is difficult to answer by itself (especially if the baseline model is not very simple).
>
> When we wrote the sentence "Varshney & Alemzadeh (2017) and Mohseni et al. (2021) suggest that directly interpretable models are an inherently safe design because humans can inspect the models to find spurious elements, but do not suggest a quantitative approach for certifying safety through interpretability as done in this work", our intention was only to contrast the more qualitative nature of the two references with the more quantitative nature of this present paper. We did not mean to be overly bold with claims about the contributions of the present paper. We have revised the sentence to read "Varshney & Alemzadeh (2017) and Mohseni et al. (2021) give qualitative accounts suggesting that directly interpretable models are an inherently safe design because humans can inspect them to find spurious elements; in this paper, we attempt to make those qualitative suggestions more quantitative."
>
> > Also, to me the proposed maximum deviation measure seems a bit conservative for evaluating the safety risk of a model. A large deviation between the prediction of two models could also mean that the new model is an improved estimator---i.e. it can model parts of the data that the old model did not model (put into the same leaf, for the example of decision trees).
>
> Please see our response to the common concern about the conservatism of maximum deviation. Moreover, we acknowledge your exact point about an improved estimator in the paragraph below eq. (1).
>
> > Last, it would be nice if the obtained score provides a guidance for domain experts to retrain a safer model in terms of the score.
> > Inspecting every subset of data points on which the new model and the baseline differ substantially might take a lot of time.
>
> In Appendix D.1, subsection titled "Feature combinations that maximize deviation," we discuss how the $\arg\max$ in (1) might inform subsequent actions, more so than the max score. (This subsection is summarized in the last paragraph of Section 5.1.) While we agree that domain expert intervention may be needed, retraining model $f$ is not the only option. One might choose the output of one of $f$, $f_0$ over the other, combine the two somehow, or abstain from prediction altogether. In the first two cases, full retraining might not be necessary if a local model editing technique is used (e.g. Daly et al., "User Driven Model Adjustment via Boolean Rule Explanations," AAAI 2020). Having said the above, formulation (1) addresses only the worst case and it may indeed take a lot of time to iterate on (1) and examine regions that successively maximize deviation. We are not sure however that this effort can or should be avoided in highly consequential applications.

---

### Official Review · Reviewer_uSea · 2021-10-27

**Correctness:** 3
**Technical Novelty And Significance:** 3
**Empirical Novelty And Significance:** 2
**Recommendation:** 5
**Confidence:** 2

**Main Review:**

### Strength
#### [Strength1]
The idea of inspecting the safety of black-box models through the approximated white-box model wills be novel and interesting.
This idea will be able to relax the difficulty of directly inspecting the safety of black-box models, where some ideal assumptions are required on the black-box models under consideration.
By putting the approximated white-box model in the middle of the inspection, the task turns into inspecting the gap between the two models, where we will need some assumptions only on the difference of the models but not the models themselves.

#### [Strength2]
The authors tried to inspect the maximum deviation under a piece-wise Lipschitzness assumptions.
In particular, the authors adopted Assumption 3 trying to characterize interpretable white-box models as piece-wise Lipschitz functions.
Although this may not always be true, nevertheless this assumption itself would be an interesting idea in the area of interpretable machine learning where interpretability is ambiguous.

### Weakness
#### [Weakness1]
The theoretical results on specific model classes in Section 4.1 and 4.2 on the deviation between decision trees and generalized additive models are straightforward and trivial.
The results on tree ensembles in Section 4.3 are less trivial, although the proposed method is a simple extension of the pioneering studies of Chen et al. (2019) and Devos et al. (2021).

I think the most important results are on Section 4.4 where we do not require specific model classes except that the models are piece-wise Lipschitz.
In practical machine learning, it is not realistic to restrict the model class to be deployed: the models can be complex deep neural networks or ensemble of several different models.
The current result in Section 4.4 is just borrowed from Bubeck et al., (2011).
I would expect to see further investigations over piece-wise Lipschitz models.
For example, with what assumptions (without specifying model classes), we can derive nontrivial better results than the general result of Bubeck et al., (2011).

#### [Weakness2]
The experimental results on Adult Income dataset (binary classification) showing that the maximum deviation on the positive class probability close to one seems to be trivial.
If the two classification models are not identical, it is not surprising that the two models have conflicted predictions for some inputs.

This observation leads to a natural question, in what circumstances we can obtain non trivial maximum deviation (e.g., the maximum deviation less than 0.1)?
For example, suppose the reference model is the depth 5 decision tree, and the approximation is the depth 3 decision tree.
Do we need the depth 5 reference model to be sufficiently close to the depth 3 approximated model to obtain the non trivial maximum deviation?
Or, is there a possibility that the maximum deviation can be small for some depth 5 reference model distinct from the depth 3 approximated model?

**Summary Of The Paper:**

In this paper, the authors proposed inspecting the deviation between the reference model (e.g., black-box model) and its approximation (e.g., white-box model).
The major motivation is on inspecting the safety of the reference black-box model through the approximated white-box model.
The safety, such as the possible maximum output within the prescribed input domain, of white-box model, such as decision tree and linear model, are easier to inspect.
Thus, if we can evaluate the maximum deviation between the black-box model and the white-box model, we can evaluate the safety of the black-box model as well.
Based on this idea, the paper considers some possible approaches for inspecting the maximum deviation between the models.
In the experiments, the authors demonstrated that the maximum deviation are considerably large in practice.

**Summary Of The Review:**

I like the idea of inspecting the safety of black-box models through approximate white-box models, and I think some ideas (Assumption 3 in particular) are interesting and novel.

However, the main theoretical results are trivial or simple modifications of the existing results.
Moreover, the experimental results showing only trivial deviations lead to a question whether it is possible to obtain non trivial maximum deviation in practice.

Overall, I think the paper is interesting in its underlying idea.
However, the technical novelty is marginal, and there remains a fundamental question when we can have non trivial maximum deviation.

---

> ### Author Response · Authors · 2021-11-16
> **Response to Reviewer uSea**
>
> **Summary of the paper**
>
> > deviation between the reference model (e.g., black-box model) and its approximation (e.g., white-box model). The major motivation is on inspecting the safety of the reference black-box model through the approximated white-box model.
>
> There may be some misunderstanding here: The model to be assessed (what the reviewer calls the "black-box model") is not the "reference" model. Rather, the reference model is what the assessed model is compared against (what the reviewer calls the "white-box model"). Note however that in general, the reference model does not have to be white-box, and it may be an existing model that was not constructed to approximate the assessed model (please see Section 2, Reference Model paragraph).
>
> **Weakness 1: Theoretical results**
>
> > The theoretical results on specific model classes in Section 4.1 and 4.2 on the deviation between decision trees and generalized additive models are straightforward and trivial. The results on tree ensembles in Section 4.3 are less trivial, although the proposed method is a simple extension of the pioneering studies of Chen et al. (2019) and Devos et al. (2021).
>
> Please see our response to the common concern about the results in Section 4 and where we think our contribution lies.
>
> > I think the most important results are on Section 4.4 where we do not require specific model classes except that the models are piece-wise Lipschitz. In practical machine learning, it is not realistic to restrict the model class to be deployed: the models can be complex deep neural networks or ensemble of several different models.
>
> We agree that depending on the domain, it may not be realistic to restrict the model class. We would like to point out however that the model classes we have covered do allow for powerful models that also retain interpretability. On the additive models front (Section 4.2), recent advances include neural additive models (NAMs) (Agarwal et al., NeurIPS 2021) and continued fraction networks (Puri et al., NeurIPS 2021) that provide competitive performance on multiple datasets and modalities. Rule-based models (Sections 4.1 and 4.3), which can be quite flexible when large (our results are not limited by the size of such models), are widely used in e.g. consumer credit and medical risk assessment (Rudin, Nature Mach. Intell. 2019), and may be strongly preferred by subject matter experts (SMEs) in domains such as manufacturing. For cases where these model classes are a viable option, a goal of our paper is to explain why users may prefer them, specifically due to the ease of assessing them for safety. We are hopeful that future work can extend the approach established in this paper to tackle safety assessment for other model classes. Please see Appendix B.5 and Appendix C, "Choice of reference model" paragraphs which discuss such extensions.
>
> > The current result in Section 4.4 is just borrowed from Bubeck et al., (2011). I would expect to see further investigations over piece-wise Lipschitz models. For example, with what assumptions (without specifying model classes), we can derive nontrivial better results than the general result of Bubeck et al., (2011).
>
> While it would be ideal if the results for this class could be improved over those in Bubeck et al. (2011), the main contribution here is formulating Assumption 3 (as you recognized), to abstract out and subsume many interpretable functions, and to allow us to find and adapt appropriate results from the bandit literature. Given the intent of the paper, we believe that better results are not a requirement for the presented results to be of value.

---

> > ### Author Response · Authors · 2021-11-16
> > **Response to Reviewer uSea continued**
> >
> > **Weakness 2: "Trivial" maximum deviations**
> >
> > > The experimental results on Adult Income dataset (binary classification) showing that the maximum deviation on the positive class probability close to one seems to be trivial.
> >
> > Please see our response to the common concern about the conservativeness of maximum deviation. In addition, we disagree with the characterization of probability differences close to one as "trivial". As discussed in Appendix D.1 (subsection titled "Feature combinations that maximize deviation"), if the points where these differences occur are not "impossible", and if each model $f$, $f_0$ seems justified in its prediction, then there is a real disagreement that raises the question of how it should be resolved.
> >
> > > If the two classification models are not identical, it is not surprising that the two models have conflicted predictions for some inputs.
> >
> > We agree but were nevertheless a bit surprised to see strong conflicts even between two simple models, both with relatively high accuracy (see page 9, first paragraph).
> >
> > > in what circumstances we can obtain non trivial maximum deviation (e.g., the maximum deviation less than 0.1)? For example, suppose the reference model is the depth 5 decision tree, and the approximation is the depth 3 decision tree. Do we need the depth 5 reference model to be sufficiently close to the depth 3 approximated model to obtain the non trivial maximum deviation? Or, is there a possibility that the maximum deviation can be small for some depth 5 reference model distinct from the depth 3 approximated model?
> >
> > Figure 1, second panel shows that a maximum deviation of 0.114 occurs when the assessed model (what the reviewer calls the "reference" model) is a 10-leaf decision tree and the reference model (what the reviewer calls the "approximated" model) is an 8-leaf DT. We have added Figure 6 in Appendix D.1, which is a second version of Figure 1 with a finer sweep over the number of leaves to see more precisely the tree sizes that result in moderate maximum deviation. Note also (going back to your earlier comment about approximation) that all decision trees are fit independently, without the explicit intent for one to be close to another. It is true that this is a case where the assessed model and reference model are from the same class. However, this would be typical in the scenario where $f$ is an updated version of a deployed model $f_0$ (see Section 2, Reference Model paragraph).

---

### Official Review · Reviewer_dzWL · 2021-11-02

**Correctness:** 2
**Technical Novelty And Significance:** 2
**Empirical Novelty And Significance:** 3
**Recommendation:** 3
**Confidence:** 4

**Main Review:**

The paper raises an important question regarding the assessment of model safety, defined by the authors as the minimax rate over a certification set, usually a superset of the input space, with respect to a safe and interpretable alternative model. However, the reviewer has some major concerns regarding the novelty and the development of the proposed methods.

Major issues:
1. The definition and motivation of "safety" could use more justification. Though the proposed minimax rate is a legit distance measure between a given model and a interpretable model, it is also a direct measure for the extrapolation behavior / generalizability / robustness, of the model, while this will trivialize the novelty of the "safety" aspect and question the necessity of defining a new term. The idea of using an interpretable function as a reference should also be better motivated and supported theoretically and empirically. It is unclear why we need a two-function approach, especially:
- The examples (existing practices / real world cases) that exercise it, which seem not available in existing literature.
- If the intended maximum deviation is large, why not directly compare to a constant function / measure the minimax.
- If the intended maximum deviation is small, what is the purpose of forcing another model to behavior almost identically w.r.t. the safe one.

2. The discussion regarding how to compute the maximum deviation can be rebalanced. The linear and the tree parts are relatively simple, whereas the more interesting piecewise Lipschitz scenario with regrets could be further extended. The derivation in its current shape leaves a couple of questions.
- The linear models and the tree models both have tractable extrapolation patterns, therefore it appears weird why to use them as f.
- The setup of a cartesian set as the certification set implies compactness, which seems to result all continuous functions to piecewise Lipschitz of any order \beta (m \to \infty when \beta \to 0). The optimal minimax rate in (14) is then likely to be dominated by the constant term overall. On the other hand, it would be useful to have examples corresponding to the functions discussed here, especially pertaining to realistic model that is discontinuous and not piece-wise constant (e.g. functional trees).

3. The empirical study is also insufficient since it only covers one dataset and one f in the main body. It therefore is hard to argue the practical value of the proposed method.



**Summary Of The Paper:**

In the paper the authors proposed the maximum deviation approach to study the safety of a predictive model by measuring its distance to an interpretable, "safe" function on a certification set. The authors explicitly showed how to compute such measure between function classes of linear, generalized additive, tree based, and piecewise Lipschitz models.

**Summary Of The Review:**

While the authors proposed a maximum deviation approach to measure model safety, the overall research question remains ambiguous and both theoretical and empirical supports for the proposed method and its claims are insufficient.

---

> ### Author Response · Authors · 2021-11-16
> **Response to Reviewer dzWL**
>
> **Major issues:**
>
> 1. **Definition and motivation of "safety"**
>
> > The definition and motivation of "safety" could use more justification. Though the proposed minimax rate is a legit distance measure between a given model and a interpretable model, it is also a direct measure for the extrapolation behavior / generalizability / robustness, of the model, while this will trivialize the novelty of the "safety" aspect and question the necessity of defining a new term.
>
> To us, "extrapolation" refers to how the output of a model outside of a region compares to its output inside the region, while (adversarial) "robustness" refers to how the output changes within a small vicinity of a point. Both of these notions involve a single function. In contrast, our notion of maximum deviation is with respect to a reference model and is thus necessarily two-function and different than extrapolation and robustness.
>
> > "minimax rate"
>
> We think this term is a mischaracterization since there is no outer minimization of the maximum deviation and no rate to speak of.
>
> > The idea of using an interpretable function as a reference should also be better motivated and supported theoretically and empirically. It is unclear why we need a two-function approach, especially:
> > - The examples (existing practices / real world cases) that exercise it, which seem not available in existing literature.
>
> For the necessity of a two-function approach, please see our previous response and Section 2, Reference Model paragraph. As for real world cases, the same Reference Model paragraph describes two cases where one has a reference model deemed to be safe: (a) simple models that can be understood by domain experts, and (b) a model that is not necessarily simple/interpretable but has been deployed extensively. Case (a) occurs for example in medical risk assessment, consumer finance, and (in our first-hand experience) predicting semiconductor yield (these examples have been added to the Reference Model paragraph). Case (b) occurs whenever one wants to replace the deployed model with a newer version.
>
> > If the intended maximum deviation is large, why not directly compare to a constant function / measure the minimax.
>
> Comparing to a constant function is a special case of our approach, but it is generally too conservative. A reference model should be allowed to vary over the input space to capture true patterns while still being considered safe.
>
> > If the intended maximum deviation is small, what is the purpose of forcing another model to behavior almost identically w.r.t. the safe one.
>
> As we wrote in Section 2 under eq. (1), we are not necessarily forcing model $f$ to behave almost identically to the safe model $f_0$. Rather, points that maximize deviation may trigger further investigation, which may then conclude that the deviation is acceptable.

---

> > ### Author Response · Authors · 2021-11-16
> > **Response to Reviewer dzWL continued**
> >
> > 2. **Questions about computing maximum deviation for model classes**
> >
> > > The discussion regarding how to compute the maximum deviation can be rebalanced. The linear and the tree parts are relatively simple, whereas the more interesting piecewise Lipschitz scenario with regrets could be further extended.
> > > The linear models and the tree models both have tractable extrapolation patterns, therefore it appears weird why to use them as f.
> >
> > Please see our response to the common concern about these results and where we think our contribution lies. In particular, while we agree that linear and tree models have tractable extrapolation patterns, we had to identify the precise properties and assumptions to state this formally.
> >
> > > The setup of a cartesian set as the certification set implies compactness, which seems to result all continuous functions to piecewise Lipschitz of any order \beta (m \to \infty when \beta \to 0). The optimal minimax rate in (14) is then likely to be dominated by the constant term overall.
> >
> > You are right in asserting that over a compact set, functions can be approximated to an arbitrary degree using piecewise Lipschitz functions. However, given that we are looking at interpretable functions where equation 14 is derived, $m_0$ and $m_1$ are likely to be small as both $f$ and $f_0$ should typically have few partitions (mentioned in the line below equation 14). In such a scenario even if $\beta$ is small, the query budget will play an important role in the bound.
> >
> > > it would be useful to have examples corresponding to the functions discussed here, especially pertaining to realistic model that is discontinuous and not piece-wise constant (e.g. functional trees).
> >
> > Examples of models that may not be piecewise constant are oblique decision trees (Murthy et al. 1994), regression trees with linear functions in the leaves and functional trees like you already suggested. We have added these examples to Section 4.4.
> >
> > 3. **Empirical study**
> >
> > > The empirical study is also insufficient since it only covers one dataset and one f in the main body. It therefore is hard to argue the practical value of the proposed method.
> >
> > Please see our common response regarding the purpose of experiments. In addition, you are probably aware that Appendix D covers a second dataset.

---

### Official Review · Reviewer_6qDo · 2021-11-03

**Correctness:** 3
**Technical Novelty And Significance:** 2
**Empirical Novelty And Significance:** 1
**Recommendation:** 6
**Confidence:** 4

**Main Review:**

**Strong points:**
* Great topic.
* Good collection of results (to be blunt: I'm not sure the results are all that surprising, but it is nonetheless good to have them written down somewhere).
* Paper easy to read.

**Weak points:**
* I think the experimental evaluation is not that convincing.  I get that it's kind of challenging to evaluate the methodology presented here, b/c (i) there seem to not be any real baselines, and (ii) it's hard to come up with an objective evaluation criteria.  But I do think there are a couple of things the authors could do to get around these issues:
    * Can you generate a synthetic data set (based on some specification of $\mathcal C$) that you essentially use as a hold out set?  I.e., evaluate both the reference and candidate model on this hold out set, and compare the two accuracies.  Now how does this number compare with the deviation you compute?  This is probably the simplest possible baseline that might still serve as "check" on your method, i.e., that it is doing something sensible.
    * You cite Wong and Kolter (2018) in your related work section.  That paper produces bounds on the robust error.  If you compare those numbers for two models, how do they compare to the deviations you get?  Perhaps you could utilize other related works in a similar way.
* As an alternative to the above, it would probably be worthwhile (and a good contribution) to carry out some kind of case study, where you run your methodology on a real-world data set, and draw out some interesting insights/findings.  (I didn't really see this being done in the appendix, but maybe I missed it.)

**Questions:** see my "Weak points" and "Additional feedback" sections.

**Additional feedback:**
* If you use the black box methodology (Sec 4.4) even when you have "closed-form" methodology (Sec 4.1) available, what happens?  How well/poorly does the black box methodology do?  Is it a bad idea to just always use the black box methodology, by default?
* To handle neural networks, there is some interesting recent work that reparametrizes a neural network in a certain way, winding up with a linear model.  I guess this could conceivably put neural networks into the more tractable part of your framework?  That seems like it could be pretty useful ...

**Summary Of The Paper:**

**Summary:**
* The paper puts forth methodology for (efficiently?) computing the worst-case deviation between two fitted models (one is a "candidate" model, the other is a "reference" model), over some feasible region $\mathcal C$ (that need not be convex).
* The takeaway message is that these kind of computations are useful for evaluating the safety of the candidate model, b/c if it (i.e., its predictions) deviate(s) too far from the reference model, then that is a bad sign and someone should "investigate something".
* Some calculations are worked out showing that for a variety of model classes, these computations can be done efficiently-ish.
* Some experimental results are presented, mainly showing how the deviation varies as the tuning parameters used to fit the candidate model are varied ...

**Summary Of The Review:**

**Recommendation:** reject.  I like the paper and topic a lot.  But it's a bit hard to objectively evaluate the paper as it is currently written.

---

> ### Author Response · Authors · 2021-11-16
> **Response to Reviewer 6qDo**
>
> **Weak points:**
>
> > evaluate both the reference and candidate model on this hold out set, and compare the two accuracies. Now how does this number compare with the deviation you compute? This is probably the simplest possible baseline that might still serve as "check" on your method, i.e., that it is doing something sensible.
>
> Thanks for this suggestion and the next one, which we interpret as partly a "sanity check" on our approach. For this suggestion, we have added Figure 7 in Appendix D.1 showing maximum deviation as a function of held-out test set accuracy for the Adult Income dataset and the DT, LR, and GAM models shown in Figure 1. Since the reference model is fixed, we did not compute the difference between the candidate model and reference model accuracies as it would just shift the values. Please take a look at Figure 7 and the accompanying paragraph in blue text. We think it does pass the sanity check and shows two regimes of interest.
>
> > You cite Wong and Kolter (2018) in your related work section. That paper produces bounds on the robust error. If you compare those numbers for two models, how do they compare to the deviations you get?
>
> We used Wong and Kolter's definition of robust loss (eq. (11) in their paper, reproduced as (16) in ours). While they focus on bounding the robust loss for feedforward neural networks, we find that our results in Section 4.2 apply to computing robust loss exactly for linear and additive models. We have thus added Figure 8 in Appendix D.1 to show the resulting robust accuracy values versus maximum deviation. Please see that figure and the accompanying blue text. It is qualitatively similar to Figure 7. We think that our machinery for decision trees also applies to computing robust loss and hope to have those results in a few days.
>
> > As an alternative to the above, it would probably be worthwhile (and a good contribution) to carry out some kind of case study, where you run your methodology on a real-world data set, and draw out some interesting insights/findings. (I didn't really see this being done in the appendix, but maybe I missed it.)
>
> We agree that a case study seems more suitable than a shallower evaluation on a large number of datasets. However, we feel that Section 5 and its continuation in Appendix D.1 already do this for the Adult Income dataset, and Appendix D.2 is a similar case study on the Lending Club dataset. We would greatly appreciate it if you could (re-)read these sections and give us specific feedback. In terms of insights:
>
> 1. We were a bit surprised that maximum deviation can be high even between two simple models, and even when the certification set radius $r$ is small (Figure 1, Figures 9-12).
> 2. For tree ensembles, while the number of trees/estimators is sometimes taken to be a measure of model complexity, we find that the maximum deviation does **not** increase as the number of estimators increases (Figure 2(a)). It may be that larger tree ensembles, while more "complex", are also smoother than smaller ones.
> 3. Feature combinations that maximize deviation ($\arg\max$ of eq. (1)): We encourage you especially to read this subsection in Appendix D.1. The main patterns were summarized in the last paragraph of Section 5.1: (a) Extrapolation of linear and monotonic functions to extreme points of $\mathcal{C}$, which while improbable do not seem *impossible*; (b) an apparent artifact in the CapitalLoss function of the GAM.

---

> > ### Author Response · Authors · 2021-11-16
> > **Response to Reviewer 6qDo continued**
> >
> > **Additional feedback:**
> >
> > > If you use the black box methodology (Sec 4.4) even when you have "closed-form" methodology (Sec 4.1) available, what happens? How well/poorly does the black box methodology do? Is it a bad idea to just always use the black box methodology, by default?
> >
> > Considering the case in Section 4.1 where both $f$ and $f_0$ are trees with $\pi$ being the number of non-empty (disjoint) intersections, then Proposition 1 states that the max deviation can be exactly computed in $\pi$ evaluations which will result in $2\pi$ queries. The result in Section 4.4 which considers both models to be interpretable would for decision trees have $\beta\rightarrow \infty$ since the function in the leaves is a constant. Then for the same number of $2\pi$ queries, we obtain an upper bound on the regret of the max deviation to be $\approx O\left(\sqrt{\frac{1}{2}}\right)$. This is likely to be a small constant; however, the closed form is still better as we get the exact value of the maximum deviation. In general, it would seem that it is better to use these closed forms, as they exploit the precise structure of the models without any slack, and resort to results in Section 4.4 only when such information is inapplicable or unavailable.
> >
> > > To handle neural networks, there is some interesting recent work that reparametrizes a neural network in a certain way, winding up with a linear model. I guess this could conceivably put neural networks into the more tractable part of your framework? That seems like it could be pretty useful ...
> >
> > This sounds interesting. Please point us to this work so that we can comment.

---

> > > ### Author Response · Authors · 2021-11-17
> > > **Robust accuracy for decision trees**
> > >
> > > We have added a plot of maximum deviation vs. robust accuracy for decision trees to Figure 8, as promised above.

---

> > > ### Comment · Reviewer_6qDo · 2021-11-29
> > > **Response to response**
> > >
> > > Re: the second comment -- sure, here are a couple of starting points:
> > >
> > > https://arxiv.org/pdf/2006.05900.pdf
> > >
> > > https://www.stat.cmu.edu/~ryantibs/papers/sparsitynn.pdf

---

> > > ### Comment · Reviewer_6qDo · 2021-11-30
> > > **Clarification**
> > >
> > > I just mean:
> > > * take a hold out set (which we can think of as a matrix wlog)
> > > * write the rows as $x_i \in \mathbb{R}^d$, $i=1,\ldots,n$
> > > * compute the two model outputs on each $x_i$, i.e., compute $f(x_i)$ and $f_0(x_i)$, for $i=1,\ldots,n$ -- provided that $x_i \in \mathcal C$
> > > * now compute $D$ at each of these $f, f_0$ pairs (so you have a bunch of $D$-values at the end of the day)
> > > * take the max of these $D$-values as the "empirical version" of your eq 1

---

> > > > ### Author Response · Authors · 2021-11-30
> > > > **We do this already (case $r = 0$)**
> > > >
> > > > We do exactly what you wrote already using the Adult Income test dataset (and Lending Club). It corresponds to ball radius $r = 0$, in which case $\mathcal{C}$ consists of only the test set and the maximum deviation can be evaluated empirically as you described (please see the last line of page 8 and the first line of page 9 in the paper). This empirical maximum deviation is in general an underestimate of the true maximum deviation when $r > 0$ and $\mathcal{C}$ is an infinite set. Figure 1, left panel shows the extent of the underestimate for LR and GAM models (for DT, the empirical deviation is already maximal). Figures 9 and 11 show breakdowns by leaves of the reference DT.

---

> > > > > ### Comment · Reviewer_6qDo · 2021-11-30
> > > > > **Clarification**
> > > > >
> > > > > Right, but my point is that it's not clear why this isn't a good enough strategy in general.  Clearly, it depends on the sample size (and possibly other characteristics of the problem).  I guess your method may still be useful when you are sample-starved.  Anyways, I can up my score a little bit (though I really think these are issues you ought to be exploring in more depth) ...

---

> > ### Comment · Reviewer_6qDo · 2021-11-29
> > **Response to response**
> >
> > First of all, thanks for your response, and for taking the time to make the additional figures!
> >
> > * Re: Fig 7 -- here is what I actually had in mind:
> >     1) compute some measure of accuracy (e.g., squared error loss) of the candidate model **over all $x \in \mathcal C$** on a hold out set;
> >     2) repeat (1) for the reference model; and
> >     3) take the max of the measure of discrepancy $D$ over the outputs of (1), (2) above.
> >
> >     This "sanity check" essentially asks: wouldn't computing the "maximum deviation" (i.e., your Eq 1) on (unseen) samples be good enough?  And if not, then under what conditions would doing so not be good enough?
> >
> > * Re: Fig 8 -- it's a little hard to tell what's going on from the plots in Fig 8, b/c you seem to have zoomed in on a very specific range of the x,y axes.  Stepping back a bit, my general feeling was: is looking at some measure of robust accuracy "good enough?"  I guess I would call a surrogate for the maximum deviation "good enough," if it was roughly correlated with the maximum deviation.  So even though there appears to be some weak correlation for high-ish (?) levels of robust accuracy, as in your figure, I would imagine the two measures are much more correlated at smaller levels of robust accuracy, and therefore maybe reasonably correlated overall (i.e., unconditionally)?
> >
> > * Re: the "insights" you mention -- maybe that first finding is surprising?  Maybe not?  Couldn't the large deviation just be explained by noise in $Y$?  It's possible there is a more interesting explanation, but it's hard to tell in the abstract ...
> >
> >     And for the second finding, I thought this could be explained by the well-known observation that boosting is "slow to overfit?"

---

> > > ### Author Response · Authors · 2021-11-30
> > > **Thanks for following up. Further questions and comments:**
> > >
> > > > Re: Fig 7 -- here is what I actually had in mind:
> > > > 1. compute some measure of accuracy (e.g., squared error loss) of the candidate model **over all** $x \in \mathcal{C}$ on a hold out set;
> > >
> > > We still do not understand what you have in mind. By "over all $x \in \mathcal{C}$", do you mean for instance the expected loss $\mathbb{E}[L(f(X), Y)]$ where $X$ follows a uniform distribution over $\mathcal{C}$? If so, we are not sure how this can be done since $Y$ is not known over all of $\mathcal{C}$ but only on a finite sample. Perhaps you are suggesting that $Y$ be synthesized by interpolating/extrapolating from its known sampled values. In that case however, we do not know of a principled way of doing it that would not overly bias the results.
> > >
> > > > Re: Fig 8 -- it's a little hard to tell what's going on from the plots in Fig 8, b/c you seem to have zoomed in on a very specific range of the x,y axes.
> > >
> > > We have not zoomed in to a specific range. The values are plotted for the same wide range of models in Figure 1 (i.e. DTs with 9-5000 leaves, the LR and GAM models listed in Tables 1 and 2).
> > >
> > > > I would call a surrogate for the maximum deviation "good enough," if it was roughly correlated with the maximum deviation. So even though there appears to be some weak correlation for high-ish (?) levels of robust accuracy, as in your figure, I would imagine the two measures are much more correlated at smaller levels of robust accuracy, and therefore maybe reasonably correlated overall (i.e., unconditionally)?
> > >
> > > Based on Figure 8, and since we have not zoomed in to relatively high levels of robust accuracy, we would conclude that robust accuracy is not a "good enough" surrogate for maximum deviation. In the LR plot, they are weakly correlated overall (if one squints), but there are distinct regimes where robust accuracy stalls while maximum deviation increases, and where robust accuracy decreases while maximum deviation remains constant. In the DT and GAM plots, the latter two regimes prevail so they are if anything anti-correlated.
> > >
> > > > maybe that first finding is surprising? Maybe not? Couldn't the large deviation just be explained by noise in
> > > Y? It's possible there is a more interesting explanation, but it's hard to tell in the abstract.
> > >
> > > Our take is that because the models are simple, noise in $Y$ is likely to be smoothed and unlikely to play a role. Rather, the models' different inductive biases may cause them to learn different approximations that, while similarly performing in an overall sense (as measured by test accuracy), yield rather different outputs in certain corners of the certification set. While we tend to think that we understand these models well, this is a reminder that small surprises can still happen.
> > >
> > > > And for the second finding, I thought this could be explained by the well-known observation that boosting is "slow to overfit?"
> > >
> > > Thanks for the suggestion. It seems like a good thing to look into and perhaps cite.

---

### Author Response · Authors · 2021-11-16
**Thanks and responses to common concerns**

Thanks to all the reviewers for your efforts and for appreciating the value of the main idea of our paper. Below we discuss common concerns raised in your reviews. We then respond to your reviews individually. Changes to the paper are highlighted in blue in the updated version.

 **Results in Section 4 do not take much work to prove (Reviewers dzWL, uSea)**

1. First, these results have not appeared before to our knowledge, especially not with two functions $f, f_0$.
2. We think Reviewer 6qDo said it well: "Good collection of results (to be blunt: I'm not sure the results are all that surprising, but it is nonetheless good to have them written down somewhere)." To elaborate on "good to have them written down," we think our contribution lies in identifying the *properties* of these interpretable models (e.g. piecewise structure, separability) and precise *assumptions* (e.g. Assumptions 1-3). These enable (a) formal mathematical statements of the fact that maximum deviation is efficiently computable, and (b) proofs of these statements that then appear straightforward.
3. A second contribution of these results is that intuitive measures of complexity for these models, such as the number of leaves and smoothness of GAM functions, are given a more precise interpretation in terms of the complexity of maximizing deviation. For example, the complexity for decision trees is bounded by the number of leaves, while the complexity for GAMs depends on the one-dimensional smoothness parameters $C_j$. We have emphasized this point at the beginning of Section 4.

**Maximum deviation is conservative (Reviewers uSea, 6m9J)**

In formulating eq. (1) as the worst case over a certification set, we made a deliberate choice to depend as little as possible on a probability distribution or a dataset sampled from one. As stated in Section 2, Certification Set paragraph, $\mathcal{C}$ can depend at most on (an expanded version of) the support set of a distribution. The reason for our choice is because safety is an out-of-distribution notion: harmful outputs often arise precisely because they were not anticipated in the data. The trade-off inherent in this choice is that the maximum deviation may be more conservative than needed. The high maximum deviation values in Figures 1, 2 may reflect this. In our view, now that definition (1) has been established in this paper, future work can consider variations that depend more on a distribution and are thus less conservative, but may also offer a weaker safety guarantee. We have added this discussion to Appendix C and respond further to Reviewer uSea and 6m9J's specific comments below.

**Purpose of experiments (Reviewers 6qDo, dzWL)**

Since our most important contribution is conceptual, our work is not one where baselines exist (as mentioned by Reviewer 6qDo) and where the purpose of experiments would be to demonstrate superiority over a large number of datasets. Instead, we agree with Reviewer 6qDo that insights from our approach are best shown through deeper case studies on one or two datasets, which we have done in Section 5 and Appendix D.

---

### Decision · Program_Chairs · 2022-01-20

**Decision:**

Reject

**Comment:**

This paper proposes a metric for the safety and interpretability of supervised learning models based on the maximum deviation from interpretable white-box models. The safety and interpretability of black-box models is an important topic, and many reviewers agree that the approach proposed by the authors is interesting. However, the maximum deviation from popular models such as decision trees, generalized linear and additive models have been intensively studied in the context of robust statistics/learning. Without explicit discussion on the connections with these existing studies, the novelty of the proposed approach cannot be properly evaluated. We thus have to conclude that the paper cannot be accepted in its current form.